# SPANNING TREE-BASED GRAPH GENERATION FOR MOLECULES

**Sungsoo Ahn**[1]**, Binghong Chen**[2]**, Tianzhe Wang**[2]**, Le Song**[3,4]
[1]POSTECH, [2]Georgia Institute of Technology, [3]Biomap, [4]MBZUAI
`sungsoo.ahn@postech.ac.kr`, {`binghong, tianzhe`}`@gatech.edu`,
`dasongle@gmail.com`

## ABSTRACT

In this paper, we explore the problem of generating molecules using deep neural networks, which has recently gained much interest in chemistry. To this end, we propose a spanning tree-based graph generation (STGG) framework based on formulating molecular graph generation as a construction of a spanning tree and the residual edges. Such a formulation exploits the sparsity of molecular graphs and allows using compact tree-constructive operations to define the molecular graph connectivity. Based on the intermediate graph structure of the construction process, our framework can constrain its generation to molecular graphs that satisfy the chemical valence rules. We also newly design a Transformer architecture with tree-based relative positional encodings for realizing the tree construction procedure. Experiments on QM9, ZINC250k, and MOSES benchmarks verify the effectiveness of the proposed framework in metrics such as validity, Fréchet ChemNet distance, and fragment similarity. We also demonstrate the usefulness of STGG in maximizing penalized LogP value of molecules.

## 1 INTRODUCTION

Researchers have extensively studied graph generative models, dating back to the early works of Erdös Rényi (Erdös et al., 1959). Recently, models based on deep neural networks (DNNs) have gained much attraction due to their expressive power in learning a graph dataset. The molecule-generating DNNs stand out among them for their success in the task of drug discovery.

Recent works have proposed molecule-generating DNNs based on string-based and graph-based representations (Segler et al., 2018; Jin et al., 2018; You et al., 2018; Shi et al., 2020; Jin et al., 2020). For example, Segler et al. (2018) proposed to train language models on the domain-specific linear string representation of molecules, i.e., simplified molecular-input line-entry system (SMILES, Weininger 1988). Since the string-based models ignore the inherent graph structure, recent works explore the graph-based generation that use (a) atom-by-atom (You et al., 2018; Shi et al., 2020; Luo et al., 2021) or (b) substructure-based (Jin et al., 2018; 2019; 2020) operations.

Notably, the substructure-based generative models (Jin et al., 2018; 2019; 2020) successfully exploit the molecular prior knowledge: the graphs are sparsely connected and can be represented as a junction tree with molecular substructure as building blocks. Based on such knowledge, the models use the junction tree construction operators which (a) require a fewer number of steps to generate the whole molecular graph and (b) guarantee generating molecules that satisfy the chemical valence rules. However, despite such advantages, a recent benchmark (Polykovskiy et al., 2020) suggests that they do not outperform the existing methods in terms of learning the data distribution, even when compared with the simple SMILES-based language models. We hypothesize that this is due to the models using a coarse-grained representation of the molecule and they may lack the ability to learn the inner semantics of each substructure-based building block.

**Contribution.** In this work, we propose a novel framework, coined spanning tree-based graph generation (STGG), for fine-grained generation of molecules while exploiting their sparsity.[1] Mainly inspired from the SMILES representation of molecules, our idea is to generate the molecular graph

---

[1]While our framework is designed for general sparse graphs, we focus on the molecular graphs in this paper.

Figure 1: (left) Benzaldehyde, (middle) its spanning tree (blue) and residual edges (red), and the corresponding constructive decisions (right). Open circle represent atoms and bonds in the molecule.

as a composition of a *spanning tree* and the corresponding *residual edges* with atoms and bonds as building blocks. Such a formulation allows our framework to utilize compact tree-constructive operations to define the molecular graph connectivity. See Figure 1 for an illustration of how we formulate the generation of a molecular graph as a sequence of tree-constructive operations.

Since our framework maintains the molecular graph structure during construction, it can pre-determine decisions that (a) violate the graph construction rule and (b) lead to molecules that violate the chemical valence rule. Such criteria allow control over the generative model to guarantee generating valid molecular graphs by forbidding invalid actions. This is in contrast to prior works (Shi et al., 2020; Luo et al., 2021) that generate the molecular graph atom-by-atom but determines the validity of construction operations through a sample-rejection scheme.

To recognize the spanning tree-based representation used in our STGG framework, we propose a Transformer architecture (Vaswani et al., 2017) with tree-based relative encoding. Inspired by recent works (Villmow et al., 2021; Lukovnikov & Fischer, 2021; Ying et al., 2021) on tree-based and graph-based Transformers, our framework expresses the relative position between two vertices as the number of forward and reverse edges in the shortest path between them. We also introduce an attention-based mechanism for constructing residual edges.

We experiment on popular graph generation benchmarks of QM9, ZINC250κ, and MOSES to validate the effectiveness of our algorithm. In the experiments on QM9 and ZINC, our STGG framework outperforms the existing graph-based generative models by a large margin. In the MOSES benchmark, our algorithm achieves superior performance compared to both string-based and graph-based methods for majority of the metrics, e.g., Fréchet ChemNet distance (Preuer et al., 2018) and fragment-based similarity. We also conduct experiments on the offline optimization task for high penalized octanol-water partition coefficient and achieve competitive results.

## 2  SPANNING TREE-BASED GENERATION OF GRAPHS (STGG)

### 2.1  OVERVIEW

In this section, we introduce our spanning tree-based graph generation (STGG) framework to sequentially generate a molecule as a composition of a spanning tree and residual edges. To this end, we propose compact tree-constructive operations inspired by the simplified molecular-input line-entry system (SMILES, Weininger, 1988). In contrast to the existing SMILES-based molecular generative methods, our framework (a) allows inferring the intermediate graph structure and (b) is generally applicable to graph types other than molecules. In particular, (a) further enables our framework to control the construction process such that the sequential operations comply with tree-constructive grammar and only generate molecules satisfying the chemical valence rule.

**Molecular graph representation.** To apply our framework, we represent a molecule as a bipartite graph $\mathcal{G} = (\mathcal{A}, \mathcal{B}, \mathcal{E})$ where $\mathcal{A}$ and $\mathcal{B}$ are the set of vertices associated with atoms and bonds of the molecule, respectively.[2] Each edge $\{a, b\} \in \mathcal{E}$ is assigned for each adjacent pair of atom and bond. We assign attributes $x_a \in \mathcal{X}_{\texttt{atom}}$ and $x_b \in \mathcal{X}_{\texttt{bond}}$ for vertices $a \in \mathcal{A}$ and $b \in \mathcal{B}$ to indicate the corresponding atom type and bond order, respectively. For example, $\{\texttt{"C"}, \texttt{"N"}, \texttt{"O"}\} \subseteq \mathcal{X}_{\texttt{atom}}$ and $\{\texttt{"-"}, \texttt{"="}\} \subseteq \mathcal{X}_{\texttt{bond}}$. See Figure 1 for an example of such a molecular graph representation.

**Molecular graph from sequence of decisions.** To generate the molecular graph $\mathcal{G} = (\mathcal{A}, \mathcal{B}, \mathcal{E})$, our framework makes a sequence of decisions $d_1, \dots, d_T$ to generate a spanning tree $\mathcal{T} = (\mathcal{A}_{\mathcal{T}}, \mathcal{B}_{\mathcal{T}}, \mathcal{E}_{\mathcal{T}})$

---

[2]Many existing works, e.g., (Shi et al., 2020), use a non-bipartite graph with bonds assigned to edges.

| attach_atom | attach_bond | branch_start | branch_end | res_atom | res_bond | terminate |
|---|---|---|---|---|---|---|
| "C" $\in \mathcal{X}_{\text{atom}}$ | "-" $\in \mathcal{X}_{\text{bond}}$ | "(" | ")" | "*" | $d \in \mathcal{L}_{\text{res}}$ | "[eos]" |

Table 1: Operations (top) and example of corresponding decisions (bottom) used in STGG.

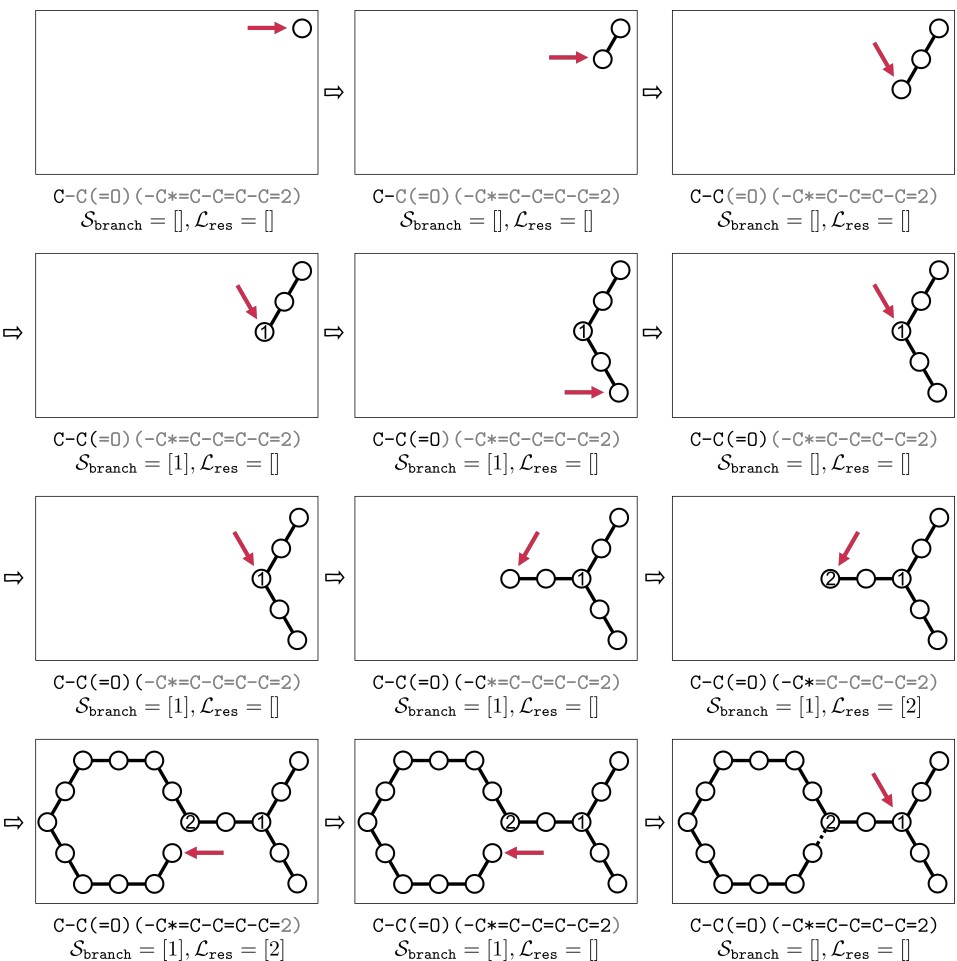

Figure 2: Demonstration of executing a sequence of decisions C-C(=O)(-C*-C-C-C-2). Here, we use the numbers 1 and 2 to mark vertices for the purpose of illustration. Next, {"C", "O"}, {"-", "="}, "(", ")", "*", and "2" correspond to attach_atom, attach_bond, branch_start, branch_end, res_atom, and res_bond operations, respectively. The decision "2" denotes res_bond operation selecting the vertex marked by "2". Decisions that are not executed at the respective time-step are faded out (gray). Location of the pointer vertex $i_{\text{point}}$ is indicated by an arrow (red).

and a set of residual edges $\mathcal{E}_R = \mathcal{E} \setminus \mathcal{E}_{\mathcal{T}}$. At each iteration, seven types of decisions are applicable, i.e., attach_atom, attach_bond, branch_start, branch_end, res_atom, res_bond, and terminate. See Table 1 for examples of decisions and the corresponding operations. We provide a detailed description of the graph construction process in Section 2.2.

**Generating valid molecular graphs.** Without any control, a model may generate decisions that (a) do not comply with the grammar of STGG or (b) leads to a molecule violating the chemical valence rule. To prevent this scenario, we conduct two criteria for determining validity of the given decision for (a) and (b). We further elaborate this in Section 2.3.

## 2.2 DECISION PROCESS FOR SPANNING TREE-BASED GRAPH GENERATION

We now explain how our STGG framework incorporates the decisions $d_1, \ldots, d_T$ to build the spanning tree $\mathcal{T} = (\mathcal{A}_{\mathcal{T}}, \mathcal{B}_{\mathcal{T}}, \mathcal{E}_{\mathcal{T}})$ and residual edges $\mathcal{E}_R$ from scratch. To this end, our framework introduces the state information of (a) a *pointer vertex* $i_{\text{point}} \in \mathcal{A}_{\mathcal{T}} \cup \mathcal{B}_{\mathcal{T}}$ for specifying the target of

---

**Algorithm 1** Tree-based generation of molecular graphs

---

1: **Input:** sequence of decisions $d_1, \ldots, d_T$.
2: **Output:** graph $\mathcal{G} = (\mathcal{A}, \mathcal{B}, \mathcal{E})$, atom attributes $\{x_a\}_{a \in \mathcal{A}}$, and bond attributes $\{x_b\}_{b \in \mathcal{B}}$
3: Set $\mathcal{A}_{\mathcal{T}} \leftarrow \emptyset, \mathcal{B}_{\mathcal{T}} \leftarrow \emptyset, \mathcal{E}_{\mathcal{T}} \leftarrow \emptyset, \mathcal{E}_R \leftarrow \emptyset$, and $\mathcal{T} \leftarrow (\mathcal{A}_{\mathcal{T}}, \mathcal{B}_{\mathcal{T}}, \mathcal{E}_{\mathcal{T}})$.      ▷ *Initialize the empty graph.*
4: Set $\mathcal{L}_{\text{res}}$ as an empty list and $\mathcal{S}_{\text{branch}}$ as an empty stack.
5: **for** $t = 1, \ldots, T$ **do**
6:      **if** $d_t \in \mathcal{X}_{\text{atom}}$ **then**      ▷ *Add a new atom vertex.*
7:          Create a new atom vertex $a$ and set $\mathcal{A} \leftarrow \mathcal{A} \cup \{b\}$ and $x_a \leftarrow d_t$.
8:          If $|\mathcal{B}_{\mathcal{T}}| > 0$, set $\mathcal{E}_{\mathcal{T}} \leftarrow \mathcal{E}_{\mathcal{T}} \cup \{\{a, i_{\text{point}}\}\}$.      ▷ *Edge is added when tree is not empty.*
9:          Set $i_{\text{point}} \leftarrow a$.
10:      **if** $d_t \in \mathcal{X}_{\text{bond}}$ **then**      ▷ *Add a new bond vertex.*
11:          Create a new bond vertex $b$ and set $\mathcal{E}_{\mathcal{T}} \leftarrow \mathcal{E}_{\mathcal{T}} \cup \{\{b, i_{\text{point}}\}\}, \mathcal{B}_{\mathcal{T}} \leftarrow \mathcal{B}_{\mathcal{T}} \cup \{b\}$, and $x_b \leftarrow d_t$.
12:          Set $i_{\text{point}} \leftarrow b$.
13:      **if** $d_t = $ "*" **then** Insert $i_{\text{point}}$ into $\mathcal{L}_{\text{res}}$.      ▷ *Add pointer vertex into the queue.*
14:      **if** $d_t \in \mathcal{L}_{\text{res}}$ **then** Pop $d_t$ from $\mathcal{L}_{\text{res}}$ and update $\mathcal{E}_R \leftarrow \mathcal{E}_R \cup \{\{i_{\text{point}}, d_t\}\}$. ▷ *Add a new residual edge.*
15:      **if** $d_t = $ "(" **then** Insert $i_{\text{point}}$ into $\mathcal{S}_{\text{branch}}$.      ▷ *Add pointer vertex into the stack.*
16:      **if** $d_t = $ ")" **then** Set $i_{\text{point}} \leftarrow \text{pop}(\mathcal{S}_{\text{branch}})$      ▷ *Update pointer vertex from the stack.*
17: Set $\mathcal{A} \leftarrow \mathcal{A}_{\mathcal{T}}, \mathcal{B} \leftarrow \mathcal{B}_{\mathcal{T}}$, and $\mathcal{E} \leftarrow \mathcal{E}_{\mathcal{T}} \cup \mathcal{E}_R$.

---

the next operation, (b) a *stack* $\mathcal{S}_{\text{branch}}$ that stores vertices to use later as a starting point of a "branch" in the spanning tree, and (c) a *list* $\mathcal{L}_{\text{res}}$ that stores vertices to use later for constructing residual edges.

In what follows, we describe the seven types of operations, i.e., `attach_atom`, `attach_bond`, `branch_start`, `branch_end`, `res_atom`, `res_bond`, and `terminate`, corresponding to decision values $d \in \mathcal{X}_{\text{atom}} \cup \mathcal{X}_{\text{bond}} \cup \mathcal{L}_{\text{res}} \cup \{$"(", ")", "*", "[eos]"$\}$ in detail. See Table 1 for the pairs of operations and the corresponding decisions. We also provide an example of the graph construction process in Figure 2.

**Attaching atom and bond vertices to the spanning tree.** If the decision $d$ specifies one of the atom or bond attributes, i.e., $d \in \mathcal{X}_{\text{atom}}$ or $d \in \mathcal{X}_{\text{bond}}$, it applies the corresponding `attach_atom` and `attach_bond` operations, respectively. To be specific, the `attach_atom` operation adds a new atom vertex $a$ into the spanning tree $\mathcal{T}$ as a neighbor of the pointer vertex $i_{\text{pointer}}$, i.e., $\mathcal{A}_{\mathcal{T}} \leftarrow \mathcal{A}_{\mathcal{T}} \cup \{a\}, \mathcal{E}_{\mathcal{T}} \leftarrow \mathcal{E}_{\mathcal{T}} \cup \{a, i_{\text{pointer}}\}$. The value $d$ is set as the new atom attribute, i.e., $x_a \leftarrow d$. The newly added vertex is set as the next pointer vertex, i.e., $i_{\text{pointer}} \leftarrow a$. The `attach_bond` operation similarly adds a new bond vertex. For example, a line graph can be expressed as a sequence of `attach_atom` and `attach_bond` operations, e.g., C-C-C where "C" $\in \mathcal{X}_{\mathcal{A}}$ and "-" $\in \mathcal{X}_{\mathcal{B}}$.

**Branching out the spanning tree.** To express graph structures with vertices of degree larger than two, our framework utilizes pairs of the `branch_start` and the `branch_end` operations with decision values of "(" and ")", respectively. To be specific, the `branch_start` operation inserts the current pointer vertex into a stack $\mathcal{S}_{\text{branch}}$ of vertices. Then the `branch_end` operation pops a vertex from the stack $\mathcal{S}_{\text{branch}}$ and sets it as the new pointer vertex. For an example, a graph with one atom vertex of degree three is constructed from a sequence of decisions, C-C(-C)(-C).

**Adding residual edges.** To construct cyclic molecular graphs, our framework generates residual edges based on pairs of `res_atom` operation and `res_bond` operation, corresponding to decision values of "*" and $d \in \mathcal{L}_{\text{res}}$, respectively. To be specific, the `res_atom` operation inserts the current (atom) pointer vertex into a list $\mathcal{L}_{\text{res}}$. Next, when the decision value $d \in \mathcal{L}_{\text{res}}$ is received for the `res_bond` operation, the corresponding vertex $d$ is popped from the list $\mathcal{L}_{\text{res}}$ and forms a new residual edge with the current (bond) pointer vertex, i.e., $\mathcal{E}_{\mathcal{R}} \leftarrow \mathcal{E}_{\mathcal{R}} \cup \{\{d, i_{\text{pointer}}\}\}$. For an example, a cyclic molecular graph is constructed from a sequence of decisions C*-O-O-1, where "1" indicates `res_bond` operation with decision of the first atom vertex with attribute "C".

**Termination.** The decision "[eos]" applies the `terminate` operation to finish the construction.

We provide the full algorithm in Algorithm 1. We also provide an algorithm to extract a sequence of decisions for constructing a given graph in Appendix A. Such an algorithm is used to obtain sequence of decisions as targets for training the generative model under the STGG framework.

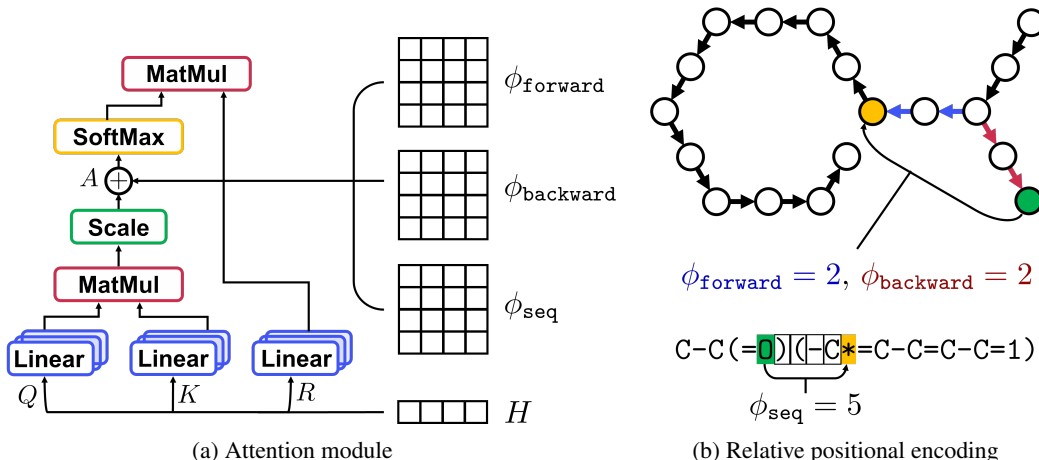

(a) Attention module                    (b) Relative positional encoding

Figure 3: Attention module and the relative positional encoding used in our framework.

### 2.3 MASKING OUT INVALID DECISIONS FOR A VALID MOLECULAR GRAPH

Based on Algorithm 1, we develop two criteria for determining whether if a sequence of decisions leads to (a) valid generation of a molecular graph and (b) generation of a molecule satisfying the valence rule. Such criteria are used to mask out invalid decisions to guarantee generating a valid molecular graph.

**Validity of graph generation.** To determine whether if a sequence of decisions lead to a valid generation of a molecular graph, we propose an algorithm that outputs a set of valid decisions given the previous decision $d$, stack of pointer vertices $\mathcal{S}_{\texttt{branch}}$, and list of atom vertices $\mathcal{L}_{\texttt{res}}$ during execution of Algorithm 1. In what follows, we provide a brief description of grammars enforced by the algorithm. We also provide the detailed algorithm in Appendix B.

- The branch_end operation only appears when the stack of pointer vertex $\mathcal{S}_{\texttt{branch}}$ is non-empty.
- The operations res_atom and res_bond are atom-specific and bond-specific, hence they only appear when the pointer vertex is located at an atom vertex and a bond vertex, respectively.
- All the bond vertices have degree of two, hence branch_start and branch_end operations only appear when the pointer vertex is located at an atom vertex.
- The stack do not contain duplicates of the a pointer vertex at the same time.

Here, we note that our criteria for valid molecular graph generation do not enforce branches and rings to be closed, e.g., C*=C-C#N is allowed by our criteria. This does not violate the validity since our Algorithm 1 may still define a valid molecular graph by ignoring the open branches and the open rings during construction, i.e., C*=C-C#N generates a molecule identical to that of C=C-C#N.

**Validity of satisfying the valence rule.** To consider chemical validity of molecules, our framework offers the ability to constrain the its generation on molecules that satisfy the *valence rule* for each atom. That is, the generated graph $\mathcal{G} = (\mathcal{A}, \mathcal{B}, \mathcal{E})$ satisfies the constraint $v(x_a) \geq \sum_{b \in \mathcal{N}(a)} o(x_b)$ for every atom vertex $a \in \mathcal{A}$ where $v(x_a)$ denotes the valence of an atom type $x_a$ and $o(x_b)$ denotes the bond order. To this end, we keep a record $r(a)$ of available valence for each atom $a \in \mathcal{A}$ and update them for each decision. For example, when a bond vertex $b$ is newly added, record of the neighboring atom vertex $a$ is updated by $r(a) \leftarrow v(a) - o(x_b)$. The main idea is to forbid actions that lead to negative values of $r(a)$. We provide a detailed algorithm in Appendix C.

## 3 TRANSFORMER ARCHITECTURE FOR TREE-BASED GENERATION

In this section, we describe our deep neural network architecture for generating sequence of decisions $d_1, \ldots, d_T$ under the STGG framework. To accurately recognize the decision process, we employ the tree-based relative positional encodings on the intermediate spanning tree $\mathcal{T}$. We also introduce an attention mechanism to express a probability distribution over $\mathcal{L}_{\texttt{res}}$ which depends on intermediate state of the algorithm.

### 3.1 TREE-BASED POSITIONAL ENCODING FOR MULTI-HEAD ATTENTION LAYERS

Each intermediate layer in our model is a combination of a multi-head self-attention module and a position-wise feed-forward neural network similar to that of Vaswani et al. (2017). The main difference is on how we modify the architecture to incorporate tree-based positional encodings. To be specific, let $H = [h_1^\top, \ldots, h_T^\top] \in \mathbb{R}^{T \times \ell}$ denote the input of a self-attention module where $d$ is the hidden dimension and $h_t \in \mathbb{R}^{1 \times \ell}$ is the hidden representation at position $t$. The input $H$ is projected by three matrices $W_Q \in \mathbb{R}^{\ell \times \ell_K}, W_K \in \mathbb{R}^{\ell \times \ell_K}$ and $W_V \in \mathbb{R}^{\ell \times \ell_V}$ to the corresponding representations $Q, K$ and $V$, respectively. A single self-attention head is then calculated as

$$Q = HW_Q, \quad K = HW_K, V = HW_V, \tag{1}$$

$$A = \frac{QK^\top}{\sqrt{\ell_K}} + P, \quad P_{t_1, t_2} = \mathbf{z}^{(1)}_{\phi_{\texttt{forward}}(t_1, t_2)} + \mathbf{z}^{(2)}_{\phi_{\texttt{backward}}(t_1, t_2)} + \mathbf{z}^{(3)}_{\phi_{\texttt{seq}}(t_1, t_2)}, \tag{2}$$

$$\text{Attention}(H) = \text{SoftMax}(M \circ A)V, \tag{3}$$

where Attention($H$) is output of the attention head, $M$ is the triangular mask to forbid the model from accessing future information while making a prediction, and $\circ$ denotes the element-wise multiplication between matrices.

Furthermore, $P$ is the newly introduced relative positional encoding. It is a summation over the trainable embedding vectors $\mathbf{z}^{(1)}, \mathbf{z}^{(2)}, \mathbf{z}^{(3)}$ indexed by relative position values of $\phi_{\texttt{forward}}(t_1, t_2), \phi_{\texttt{backward}}(t_1, t_2)$, and $\phi_{\texttt{seq}}(t_1, t_2)$. To be specific, the *tree-based relative positions* $\phi_{\texttt{forward}}(t_1, t_2)$ and $\phi_{\texttt{backward}}(t_1, t_2)$ denotes the number of forward and backward edges in the spanning tree path between pointer vertices at the $t_1$-th and $t_2$-th time step. The direction of edge is decided by order of generation in the STGG framework. Such an encoding was inspired from recent works (Villmow et al., 2021; Lukovnikov & Fischer, 2021; Ying et al., 2021) using Transformers to recognize graphs and trees. Finally, the *sequence-based relative position* $\phi_{\texttt{seq}}(t_1, t_2) = t_1 - t_2$ denotes the relative difference of time-steps for the decisions.

### 3.2 ATTENTION FOR UPDATING RESIDUAL EDGES.

Our model generates a categorical distribution over the space of $\mathcal{X} = \mathcal{X}_{\texttt{atom}} \cup \mathcal{X}_{\texttt{bond}} \cup \mathcal{L}_{\texttt{res}} \cup \{\texttt{"("},\texttt{")"},\texttt{"*"},\texttt{"[eos]"}\}$. It is relatively straight-forward to output an unnormalized probability over values of $\mathcal{X}_{\texttt{atom}} \cup \mathcal{X}_{\texttt{bond}} \cup \{\texttt{"("},\texttt{")"},\texttt{"*"}\}$ using a linear classifier on top of the Transformer model. However, it is non-trivial to assign probability values for res_bond operation, i.e., decisions values of $d \in \mathcal{L}_{\texttt{res}}$, since $\mathcal{L}_{\texttt{res}}$ varies between different time-steps. To handle this case, we use an attention-based mechanism for assigning unnormalized probability to decision values in the list $\mathcal{L}_{\texttt{res}}$. To be specific, at the final layer of our model, we obtain the following probability distribution $p(d)$.

$$p(d) \propto \begin{cases} m_g(d) \cdot m_v(d) \cdot \exp(\mathbf{w}_d^\top h) & \forall d \in \mathcal{X}_{\texttt{atom}} \cup \mathcal{X}_{\texttt{bond}} \cup \{\texttt{"("},\texttt{")"},\texttt{"*"}\}, \\ m_g(d) \cdot m_v(d) \cdot \exp(h_d^\top W_1 W_2^\top h) & \forall d \in \mathcal{L}_{\texttt{res}}, \end{cases} \tag{4}$$

where $\mathbf{w}_d \in \mathbb{R}^{1 \times \ell}$ is a decision-specific vector, $W_1, W_2 \in \mathbb{R}^{\ell \times \tilde{\ell}}$ are weight matrices, and $h$ is the decision embedding, i.e., output of the Transformer layer corresponding to the previously made decision. Furthermore, $h_d$ is the embedding corresponding to a past decision $d \in \mathcal{L}_{\texttt{res}}$. Finally, $m_v(d), m_g(d)$ are the mask for excluding invalid decisions that violate the validity of graph generation and valence rule, respectively. The masks are obtained using the criteria explained in Section 2.3. We use the mask during both training and evaluation of the model; this differs from existing graph-generative models which forbid invalid decisions only at evaluation using a sample-rejection scheme.

## 4 RELATED WORKS

**SMILES-based molecular generative models.** Several studies proposed to generate a SMILES representation of molecules using string-based (Gómez-Bombarelli et al., 2016; Segler et al., 2018; Kim et al., 2021) or grammar-based (Kusner et al., 2017; Dai et al., 2018) models. While our STGG is largely inspired from such works, our STGG allows realizing the intermediate graph structure of the molecule being constructed while the SMILES-based models cannot. This difference allows the adoption of structure-aware deep neural networks to STGG. To be specific, the difference between STGG and the SMILES-based models appears from our newly introduced graph construction procedure using a pointer vertex $i_{\texttt{point}}$, a vertex-list $\mathcal{L}$, and a vertex-stack $\mathcal{S}$. They allow recognizing an incomplete sequence of decisions as a graph and assigning positions to each decision. In contrast,

Table 2: Experimental results on ZINC250K and QM9 datasets.

| METHOD | CORRECTABLE | ZINC250K | | | QM9 | | |
|---|---|---|---|---|---|---|---|
| | | VALID | UNIQUE | NOVEL | VALID | UNIQUE | NOVEL |
| GCPN (You et al., 2018) | ✓ | 0.20 | **1.0000** | **1.0000** | - | - | - |
| MRNN (Popova et al., 2019) | ✓ | 0.65 | 0.9989 | **1.0000** | - | - | - |
| GRAPHNVP (Madhawa et al., 2019) | | 0.426 | 0.948 | **1.0000** | 0.831 | **0.992** | 0.582 |
| GRF (Honda et al., 2019) | | 0.734 | 0.537 | **1.0000** | 0.845 | 0.66 | 0.586 |
| GRAPHAF (Shi et al., 2020) | ✓ | 0.680 | 0.991 | **1.0000** | 0.67 | 0.9415 | 0.8883 |
| MOFLOW (Zang & Wang, 2020) | ✓ | 0.680 | 0.991 | **1.0000** | 0.8896 | 0.9853 | 0.9604 |
| GRAPHCNF (Lippe & Gavves, 2021) | | 0.9635 | 0.9998 | 0.9998 | - | - | - |
| GRAPHDF (Luo et al., 2021) | ✓ | 0.8903 | 0.9916 | **1.0000** | 0.8267 | 0.9762 | **0.9810** |
| SMILES-TRANSFORMER | | 0.9558 | 0.9998 | 0.9946 | 0.9908 | 0.9629 | 0.6939 |
| STGG (ours) | ✓ | **0.9950** | 0.9999 | 0.9989 | **1.0000** | 0.9676 | 0.7273 |

an incomplete SMILES string does not define a graph structure and assigning positions to each character is non-trivial.

**Graph-based molecular generative models.** Researchers have developed a large variety of molecular graph generation frameworks based on atom-wise and bond-wise operations (You et al., 2018; Kajino, 2019; Popova et al., 2019; Madhawa et al., 2019; Honda et al., 2019; Shi et al., 2020; Zang & Wang, 2020; Luo et al., 2021). Our STGG framework simplifies the decision space of such models by exploiting the tree-like graph structures of molecules. To be specific, STGG requires $O(|\mathcal{A}| + |\mathcal{B}|)$ decisions for constructing a molecule while the existing atom-by-atom graph generative models typically require $O(|\mathcal{A}|^2)$ decisions. This implies that our generative model requires a smaller number of decisions for sparse graphs like molecules, i.e., when $|\mathcal{B}|$ is small. Furthermore, our work is the first to successfully train a Transformer architecture (Vaswani et al., 2017) for graph-based molecule generation.

In another line of research, several works (Jin et al., 2018; 2019; 2020) proposed generative models based on using the junction-tree representation with molecular substructures as building blocks. Based on such a representation, such works utilize tree-constructive operations to generate the full graph. Since they operate on such a coarse-grained molecular representation, they typically require a fewer number of building blocks to generate the whole molecule. In comparison, our STGG framework utilizes a more fine-grained molecular representation and may additionally learn the inner semantics of substructures that are used as building blocks for the junction tree.

# 5 EXPERIMENT

In this section, we report the experimental results of the proposed spanning tree-based graph generation (STGG) framework. In Section 5.1 and 5.2, we compare with the existing graph generative models in the ZINC250K (Irwin et al., 2012) and QM9 (Ramakrishnan et al., 2014). We provide ablation studies on each component of our method using the ZINC250K dataset. In Section 5.2, we compare with the existing molecule generative models using the MOSES benchmark (Polykovskiy et al., 2020). Finally, in Section 5.3, we provide our results on the molecular optimization task with respect to the penalized octanol-water partition coefficient function (PLOGP). We provide the implementation details and illustration of the generated molecules in Appendix D and E, respectively.

## 5.1 MOLECULE GENERATION ON ZINC250K AND QM9 DATASETS

We first compare to the literature standard for the molecular generation task in the ZINC250K and the QM9 datasets. To this end, we train our generative model on the respective datasets and sample 10,000 molecules to measure (a) the ratio of valid molecules (VALID), (b) the ratio of unique molecules (UNIQUE), and (c) the ratio of novel molecules with respect to the training dataset (NOVEL). We compare with the numbers reported by recently proposed graph generative models (Shi et al., 2020; Luo et al., 2021). We also provide an additional baseline of a transformer architecture trained to generate the SMILES representation for the molecule (SMILES-TRANSFORMER).

We mark CORRECTABLE for methods which can optionally use a sample-rejection scheme to forbid decisions that violate the chemical rules at evaluation. Note that our framework can train the generative model under the valence correction mask *during training*, while existing graph generative

Table 4: First set of experimental results for the MOSES benchmark.

| METHOD | VALID | UNIQUE1 | UNIQUE2 | INTDIV | INTDIV2 | FILTERS | NOVEL |
|---|---|---|---|---|---|---|---|
| TRAINING DATASET | 1.0000 | 1.0000 | 1.0000 | 0.8567 | 0.8508 | 1.0000 | 0.0000 |
| HMM (Polykovskiy et al., 2020) | 0.0760 | 0.6230 | 0.5671 | 0.8466 | 0.8104 | 0.9024 | **0.9994** |
| NGRAM (Polykovskiy et al., 2020) | 0.2376 | 0.9740 | 0.9217 | **0.8738** | **0.8644** | 0.9582 | 0.9694 |
| COMBINATORIAL (Liu et al., 2017) | **1.0000** | 0.9983 | 0.9909 | 0.8732 | 0.8666 | 0.9557 | 0.9878 |
| CHARRNN (Segler et al., 2018) | 0.9748 | **1.0000** | 0.9994 | 0.8562 | 0.8503 | 0.9943 | 0.8419 |
| AAE (Polykovskiy et al., 2020) | 0.9368 | **1.0000** | 0.9994 | 0.8557 | 0.8499 | 0.9960 | 0.7931 |
| VAE (Polykovskiy et al., 2020) | 0.9767 | **1.0000** | 0.9984 | 0.8558 | 0.8498 | 0.9970 | 0.6949 |
| JT-VAE (Jin et al., 2018) | **1.0000** | **1.0000** | 0.9996 | 0.8551 | 0.8493 | 0.9760 | 0.9143 |
| LATENTGAN (Prykhodko et al., 2019) | 0.8966 | **1.0000** | 0.9968 | 0.8565 | 0.8505 | 0.9735 | 0.9498 |
| STGG (ours) | **1.0000** | **1.0000** | 0.9987 | 0.8556 | 0.8496 | **0.9976** | 0.6727 |

Table 5: Second set of experimental results for the MOSES benchmark.

| | FCD (↓) | | SNN (↑) | | FRAG (↑) | | SCAF (↑) | |
|---|---|---|---|---|---|---|---|---|
| METHOD | TEST | TESTSF | TEST | TESTSF | TEST | TESTSF | TEST | TESTSF |
| TRAINING DATASET | 0.0080 | 0.4755 | 0.6419 | 0.5859 | 1.0000 | 0.9986 | 0.9907 | 0.0000 |
| HMM (Polykovskiy et al., 2020) | 24.466 | 25.431 | 0.3876 | 0.3795 | 0.5754 | 0.5681 | 0.2065 | 0.0490 |
| NGRAM (Polykovskiy et al., 2020) | 5.5069 | 6.2306 | 0.5209 | 0.4997 | 0.9846 | 0.9815 | 0.5302 | 0.0977 |
| COMBINATORIAL (Liu et al., 2017) | 4.2375 | 4.5113 | 0.4514 | 0.4388 | 0.9912 | 0.9904 | 0.4445 | 0.0865 |
| CHARRNN (Segler et al., 2018) | 0.0732 | 0.5204 | 0.6015 | 0.5649 | **0.9998** | 0.9983 | 0.9242 | **0.1101** |
| AAE (Polykovskiy et al., 2020) | 0.5555 | 1.0572 | 0.6081 | 0.5677 | 0.9910 | 0.9905 | 0.9022 | 0.0789 |
| VAE (Polykovskiy et al., 2020) | 0.0990 | 0.5670 | 0.6257 | 0.5783 | 0.9994 | 0.9984 | 0.9386 | 0.0588 |
| JT-VAE (Jin et al., 2018) | 0.3954 | 0.9382 | 0.5477 | 0.5194 | 0.9965 | 0.9947 | 0.8964 | 0.1009 |
| LATENTGAN (Prykhodko et al., 2019) | 0.2968 | 0.8281 | 0.5371 | 0.5132 | 0.9986 | 0.9972 | 0.8867 | 0.1072 |
| STGG (ours) | **0.0680** | **0.5032** | **0.6359** | **0.5851** | **0.9998** | **0.9984** | **0.9416** | 0.0389 |

models use the valence correction *only at evaluation*. However, for comparison, we do not use the valence correction mask during training in this experiment.

We report the experimental results in Table 2. In the table, we observe that our STGG framework outperforms all the existing molecular graph generative models for high VALID at the cost of relatively lower NOVEL. In particular, our generative model can achieve a $100\%$ ratio of valid molecules in the QM9 dataset even without any correction procedure. Such a result highlights the how our model can effectively learn the chemical rules and model the underlying distribution.

Table 3: Ablation on ZINC250K.

| Method | VALID | UNIQUE | NOVEL |
|---|---|---|---|
| A | 0.9296 | 0.9995 | 0.9984 |
| S | 0.9814 | 0.9995 | 0.9955 |
| S+T | 0.9834 | 0.9997 | 0.9961 |
| S+T+G | 0.9950 | **0.9999** | **0.9989** |
| S+T+G+V | **1.0000** | 0.9992 | 0.9927 |

Finally, our STGG framework performing better than the SMILES-based transformer implies how the performance of our generative model stems from the STGG framework, rather than using the Transformer architecture.

**Ablation studies.** We also conduct ablation studies on the ZINC250k dataset to verify the effectiveness of our method. To this end, we report the experimental results of our method without specific components. To be specific, we ablate the effects of using sequential relative positional encoding (S), tree-based relative positional encoding (T), graph-construction mask (G), and valence rule mask (V). We also consider an additional baseline of using the absolute positional encoding (A) as in the original Transformer architecture (Vaswani et al., 2017). In Table 3 and Figure 4, one can observe how each component of our algorithm is crucial for achieving high VALID. In particular, the tree encoding is essential for the performance, showing the importance of tree-based representation that we use in our model.

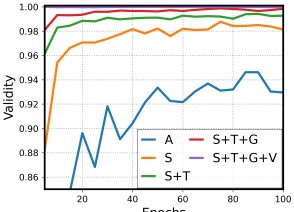

Figure 4: Ablation on ZINC250K.

### 5.2 MOLECULE GENERATION ON THE MOSES BENCHMARK

We also compare our method on the MOSES benchmark with the existing models. The MOSES benchmark offers a large collection of metrics to access the overall quality of generated molecules. To be specific, in addition to VALID, UNIQUE, NOVEL, we consider internal diversity of molecules (INTDIV), ratio of samples being accepted to chemical filters (FILTERS), Frétchet ChemNet Distance (FCD), nearest neighborhood similarity (SNN), frament similarity (FRAG), and Scaffold similarity

Figure 5: Molecular optimization results with the top-3
property scores denoted by 1ST, 2ND, and 3RD.

| Method | OFFLINE | PLOGP | | |
|---|---|---|---|---|
| | | 1ST | 2ND | 3RD |
| GVAE (Kusner et al., 2017) | ✓ | 2.94 | 2.89 | 2.80 |
| SD-VAE (Dai et al., 2018) | ✓ | 4.04 | 3.50 | 2.96 |
| JT-VAE (Jin et al., 2018) | | 5.30 | 4.93 | 4.49 |
| MHG-VAE (Kajino, 2019) | | 5.56 | 5.40 | 5.34 |
| GRAPHAF (Shi et al., 2020) | | 12.23 | 11.29 | 11.05 |
| GRAPHDF (Luo et al., 2021) | | 13.70 | 13.18 | 13.17 |
| STGG, $\gamma = 4$ (ours) | ✓ | 4.56 | 4.55 | 4.53 |
| STGG, $\gamma = 5$ (ours) | ✓ | 5.06 | 4.89 | 4.86 |
| STGG, $\gamma = 6$ (ours) | ✓ | 5.72 | 5.15 | 4.92 |
| STGG, $\gamma = 7$ (ours) | ✓ | **23.32** | **18.75** | **16.50** |

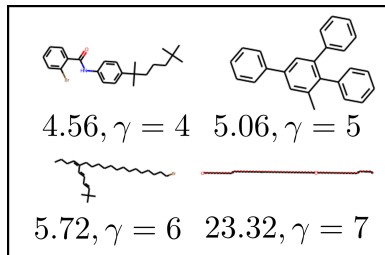

$4.56, \gamma = 4$    $5.06, \gamma = 5$

$5.72, \gamma = 6$    $23.32, \gamma = 7$

Figure 6: Optimized 1ST molecules

(SCAF). The similarity metrics of FCD, SNN, FRAG, SCAF are measured with respect to the test dataset of molecules and the scaffolds extracted from them.

In Table 4 and 5, we provide our experimental result. Here, one can observe how our algorithm outperforms the existing works for 10 out of 15 metrics including FILTERS, FCD-TEST, FCD-TESTSF, SNN-TEST, SNN-TESTSF, FRAG-TEST, FRAG-TESTSF, and SCAF-TEST. This highlights the ability of our STGG framework to successfully learn the training distribution.

### 5.3 MOLECULAR OPTIMIZATION FOR PENALIZED OCTANOL-WATER PARTITION COEFFICIENT

Finally, we demonstrate the usefulness of our STGG framework for the task of molecular optimization. To this end, we consider the literature standard of maximizing the penalized octanol-water partition coefficient (PLOGP). However, several works (Gao & Coley, 2020; Coley, 2020) have noted how the existing algorithms on this benchmark may not be practical, since PLOGP is ill-defined as a scoring function for molecules; this scoring function may assign high values to "unrealistic" molecules that are unstable and hard to synthesize in practice.

To consider this aspect, we propose a new algorithm which can control the quality of molecules by trading off scores and realistic-ness of molecules. Using this algorithm, we demonstrate how our STGG is capable of generating both (a) high-scoring molecules and (b) realistic molecules with a reasonably high score. At a high-level, we train a conditional generative model $p_\theta(m|\gamma)$ under the STGG framework with PLOGP as the condition $\gamma$. At the test time, we sample from a high value $\gamma$ to obtain high-scoring molecules. Such an algorithm is inspired from the recent offline reinforcement learning algorithms (Schmidhuber, 2019; Kumar & Levine, 2020; Chen et al., 2021; Janner et al., 2021). We fully describe our molecular optimization algorithm in Appendix F.

In Table 5 and Figure 6, we report the result of our molecular optimization experiment. We provide additional illustrations of the generated molecules in Appendix G. Here, our STGG model is able to generate molecules with considerably high PLOGP scores outside the training distribution. Furthermore, in Figure 6 and Appendix G, one can observe how increasing $\gamma$ gradually changes the optimized molecule from realistic structures to large, chain-like, and unrealistic structures.[3] Given such results, one may conclude that our STGG combined with the offline optimization algorithm can successfully make a trade-off between high PLOGP and realistic-ness of the generated molecules. However, we also remark that our results do not imply our optimization results to be strictly better than the baselines; we believe it is necessary to develop and incorporate quantitative measures for realistic-ness of molecules to fairly evaluate the molecular optimization algorithms. We believe such a research to be a important future direction.

## 6 CONCLUSION

In this paper, we propose STGG which is the first spanning tree-based framework for the generation of molecules using the Transformer architecture. The key idea of using the spanning tree for graph generation applies to any graph type outside the molecules; we believe such an extension of our work to be both promising and interesting. We also propose an offline algorithm for molecular optimization which allows the trade-off between the high score and the realistic-ness of molecules. We leave more investigation of the newly proposed optimization algorithm as future work.

---

[3]This is in agreement with prior works (Shi et al., 2020; Ahn et al., 2020; Luo et al., 2021).

## 7 REPRODUCIBILITY STATEMENT

We provide explicit description of our algorithm in Algorithm 1, Appendix 2, 3, and 4. We list the hyper-parameters, the hardware used for the experiments, and the data-processing information in Appendix D. We provide illustrations of the molecules generated for the experiments in Figure 6, and Appendix E, G. We submit the full implementation of our STGG framework and the baselines used in our experiments as a supplementary material.

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

# A EXTRACTING SEQUENCE OF DECISIONS FROM A MOLECULAR GRAPH

In this section, we explain our algorithm for finding a sequence of decisions to construct a given molecular graph $\mathcal{G} = (\mathcal{A}, \mathcal{B}, \mathcal{E})$. The high-level idea is to first perform a depth-first search on $\mathcal{G}$ to find a spanning tree $\mathcal{T} = (\mathcal{A}, \mathcal{B}, \mathcal{E}_{\mathcal{T}})$ and the corresponding set of residual edges $\mathcal{E}_R = \mathcal{E} \backslash \mathcal{E}_T$. Then the algorithm traverses the spanning tree $\mathcal{T}$ according to the depth-first search tree while (a) allocating `branch_start` and `branch_end` for vertices with degree higher than two and (b) adding `res_atom` and `res_bond` operations for any vertex covered by a residual edge $\{a, b\} \in \mathcal{E}_R$.

To this end, we utilize a stack $\mathcal{S}_{\text{dfs}}$ that stores the list of vertices in $\mathcal{G}$ and branching tokens $\{$ `"("`, `")"` $\}$ to visit. At each iteration, an element $i$ of the stack $\mathcal{S}_{\text{dfs}}$ is popped. If $i$ is a vertex, the algorithm adds the corresponding decision for `attach_atom` and `attach_bond` operations. If the vertex has more than two successors with respect to the spanning tree $\mathcal{T}$, the successors are inserted into the stack $\mathcal{S}_{\text{dfs}}$ with surrounding `"("` and `")"` tokens. If the vertex has only one successor, the successor is inserted into the stack without an additional operation. When the branching tokens $\{$ `"("`, `")"` $\}$ are popped from the stack, the algorithm adds the corresponding decision value to the sequence of decisions. We describe the full scheme in Algorithm 2.

---

**Algorithm 2** Generating sequence of decisions for a molecular graphs

---

1: **Input:** graph $\mathcal{G} = (\mathcal{A}, \mathcal{B}, \mathcal{E})$, atom attributes $\{x_a\}_{a \in \mathcal{A}}$, and bond attributes $\{x_b\}_{b \in \mathcal{B}}$
2: Find a spanning tree $\mathcal{T} = (\mathcal{A}, \mathcal{B}, \mathcal{E}_{\mathcal{T}})$ of $\mathcal{G}$ based on depth-first order and set $\mathcal{E}_R \leftarrow \mathcal{E} \setminus \mathcal{E}_{\mathcal{T}}$.
3: Initialize an empty sequence of decisions $\mathcal{D}$.
4: Choose the root $a \in \mathcal{A}$ of $\mathcal{T}$ to insert in an empty stack $\mathcal{S}_{\text{branch}}$.
5: **do**
6:      Pop $i$ from $\mathcal{S}_{\text{branch}}$.
7:      **if** $i \in \mathcal{A} \cup \mathcal{B}$ **then**
8:          Append $x_i$ to $\mathcal{D}$.                 $\triangleright$ *Decision to attach the atom vertex.*
9:          **for** $j \in \{j | j \in \mathcal{N}(i), \{i, j\} \in \mathcal{E}_{\mathcal{R}}\}$ **do**        $\triangleright$ *Decisions for residual edges.*
10:             If $i \in \mathcal{A}$, append `"*"` to $\mathcal{D}$.
11:             If $i \in \mathcal{B}$, append $j$ to $\mathcal{D}$.
12:          Let $\mathcal{V}$ denote $\{j | j \in \mathcal{N}(i), j \notin \mathcal{A}_{\mathcal{T}} \cup \mathcal{B}_{\mathcal{T}}\}$.      $\triangleright$ *Successors of $i$ in depth-first order.*
13:          If $|\mathcal{V}| > 1$, insert `"("` to $\mathcal{S}_{\text{branch}}$.      $\triangleright$ *Allocate decision to record pointer vertex.*
14:          Insert vertices in $\mathcal{V}$ to $\mathcal{S}_{\text{branch}}$.             $\triangleright$ *Allocate successors to visit later.*
15:          If $|\mathcal{V}| > 1$, insert `")"` to $\mathcal{S}_{\text{branch}}$.     $\triangleright$ *Allocate decision to return to the pointer vertex.*
16:      **if** $i \in \{$ `"("`, `")"` $\}$ **then**
17:          Append $i$ to $\mathcal{D}$.
18: **while** $|\mathcal{S}_{\text{branch}}| > 0$
19: **Output:** sequence of decisions $\mathcal{D} = d_1, \ldots, d_T$ to reconstruct $\mathcal{G}$.

---

# B  ALGORITHMS FOR GRAPH MASKING

To determine whether if a sequence of decisions lead to a valid generation of a molecular graph, we propose an algorithm that outputs a set of valid decisions given the current decision $d$, stack of pointer vertices $\mathcal{S}$, and list of atom vertices $\mathcal{L}$ during execution of Algorithhm 1. We provide the full description in Algorithm 3.

---

**Algorithm 3** Determination of grammar violation

---

1: **Input:** current decision $d$, stack $\mathcal{S}_{\text{branch}}$, and list $\mathcal{L}_{\text{res}}$.
2: **Output:** List of candidate decisions $\mathcal{D}$ that are valid.
3: **if** $d \in \mathcal{X}_{\text{atom}}$ **then**
4:     Set $\mathcal{D} \leftarrow \mathcal{X}_{\text{bond}}$.                                  ▷ *When the atom vertex is followed by bond vertex.*
5:     Set $\mathcal{D} \leftarrow \mathcal{D} \cup \{\texttt{"("}\}$                          ▷ *When the atom vertex has more than one successors.*
6:     Set $\mathcal{D} \leftarrow \mathcal{D} \cup \{\texttt{"*"}\}$                          ▷ *When the atom vertex has a neighboring residual edge.*
7:     If $|\mathcal{S}_{\text{branch}}| > 0$, set $\mathcal{D} \leftarrow \mathcal{D} \cup \{\texttt{")"}\}$.        ▷ *The "$)$" decision appears only when $\mathcal{S}_{\text{branch}}$ is non-empty*
8:     Set $\mathcal{D} \leftarrow \mathcal{D} \cup \{\texttt{"[eos]"}\}$.                          ▷ *Allow termination.*
9: **if** $d \in \mathcal{X}_{\text{bond}}$ **then**
10:     Set $\mathcal{D} \leftarrow \mathcal{X}_{\text{atom}}$.                                  ▷ *When the bond vertex is followed by atom vertex.*
11:     Set $\mathcal{D} \leftarrow \mathcal{D} \cup \mathcal{L}_{\text{res}}$                          ▷ *When the bond vertex has a neighboring residual edge.*
12: **if** $d = \texttt{"*"}$ **then**
13:     Set $\mathcal{D} \leftarrow \mathcal{X}_{\text{bond}}$.                                  ▷ *When the atom vertex is followed by bond vertex.*
14:     Set $\mathcal{D} \leftarrow \mathcal{D} \cup \{\texttt{"("}\}$                          ▷ *When the atom vertex has more than one successors.*
15:     Set $\mathcal{D} \leftarrow \mathcal{D} \cup \{\texttt{"*"}\}$                          ▷ *When the atom vertex has a neighboring residual edge.*
16:     If $|\mathcal{S}_{\text{branch}}| > 0$, set $\mathcal{D} \leftarrow \mathcal{D} \cup \{\texttt{")"}\}$.        ▷ *The "$)$" decision appears only when $\mathcal{S}_{\text{branch}}$ is non-empty*
17:     Set $\mathcal{D} \leftarrow \mathcal{D} \cup \{\texttt{"[eos]"}\}$.                          ▷ *Allow termination.*
18: **if** $d \in \mathcal{L}_{\text{res}}$ **then**
19:     Set $\mathcal{D} \leftarrow \{\}$.                                  ▷ *Residual edge is only constructed at end of each branch.*
20: **if** $d = \texttt{"("}$ **then**
21:     Set $\mathcal{D} \leftarrow \mathcal{X}_{\text{bond}}$.                                  ▷ *Branch always starts with a bond vertex.*
22:     Set $\mathcal{D} \leftarrow \mathcal{D} \cup \{\texttt{"[eos]"}\}$.                          ▷ *Allow termination.*
23: **if** $d = \texttt{")"}$ **then**
24:     Set $\mathcal{D} \leftarrow \mathcal{X}_{\text{bond}} \cup \{\texttt{"("}, \texttt{")"}\}$.        ▷ *Branch is followed by start or end of another branch.*
25:     Set $\mathcal{D} \leftarrow \mathcal{D} \cup \{\texttt{"[eos]"}\}$.                          ▷ *Allow termination.*

---

We also establish theoretical result on how the sequence of decisions generated from Algorithm 3 is always a valid sequence of decisions for Algorithm 1. To this end, we define a valid molecular graph as follows.

**Definition 1.** *A valid molecular graph* $\mathcal{G} = (\mathcal{A}, \mathcal{B}, \mathcal{E})$ *is a connected bipartite graph where the number of vertices adjacent to any bond vertex* $b \in \mathcal{B}$ *is exactly two, i.e.,* $|\mathcal{N}(b)| = 2$.

Such a definition implies how a molecule should have exactly two atoms connected to a bond. Combined with additional conditions to guarantee the well-behavior of Algorithm 1 on sequence of decisions, we obtain the following result.

**Theorem 1.** *Let* $\mathcal{G} = (\mathcal{A}, \mathcal{B}, \mathcal{E})$, $\mathcal{S}$, *and* $\mathcal{L}$ *be a graph, a stack of vertices, and a list of vertices being updated by Algorithm 1 and a sequence of decisions* $d_1, \ldots, d_T$. *If the sequence of decisions satisfies the criteria defined by Algorithm 3, the following properties are satisfied.*

$\mathcal{P}1$ *At the* $t$-*th step of Algorithm 1,* $|\mathcal{S}| > 0$ *if* $d_t = \texttt{")"}$.

$\mathcal{P}2$ *At the* $t$-*th step of Algorithm 1,* $d_t \in \mathcal{L}$ *if* $d_t \in \mathcal{A} \cup \mathcal{B}$.

$\mathcal{P}3$ *When* $d_T = \texttt{[eos]}$, *the graph* $\mathcal{G}$ *is a valid molecular graph.*

*Here,* $\mathcal{P}1$ *and* $\mathcal{P}2$ *implies how the operations in Algorithm 1 are well-defined for* $d_1, \ldots, d_T$.

*Proof.* First, $\mathcal{P}1$ is enforced by the step in Algorithm 3 which forbids the decision value of `")"` when the stack $\mathcal{S}$ is empty. Next, $\mathcal{P}2$ is enforced by the step selecting decision values from the current list of vertices $\mathcal{L}$.

To enforce $\mathcal{P}3$, when $d_T = $ `[eos]`, $\mathcal{G}$ (a) has to be a connected bipartite graph and (b) the number of vertices adjacent to any bond vertex has to be exactly two. For (a), Algorithm 3 allows the decision of $d_t \in \mathcal{X}_{\text{atom}} \cup \mathcal{L}$ only when the pointer vertex is a bond vertex, i.e., $d \in \mathcal{X}_{\text{bond}}$. Similarly, $d_t \in \mathcal{X}_{\text{bond}}$ is allowed only when $d \in \mathcal{X}_{\text{atom}} \cup \{$`"*"`, `"("`$\}$. For (b), the algorithm does not allow adding a bond vertex $b \in \mathcal{B}$ to the list of vertices $\mathcal{L}$, which is required for any vertex with degree higher than two. Termination is not allowed when there exists a bond vertex with degree smaller than two.

$\square$

# C   ALGORITHM FOR VALENCE MASKING

To consider chemical validity of molecules, our framework offers the ability to constrain its generation on molecules that satisfy the *valence rule* for each atom. That is, the generated graph $\mathcal{G} = (\mathcal{A}, \mathcal{B}, \mathcal{E})$ satisfies the constraint $v(x_a) \geq \sum_{b \in \mathcal{N}(a)} o(x_b)$ for every atom vertex $a \in \mathcal{A}$ where $v(x_a)$ denotes the valence of an atom type $x_a$ and $o(x_b)$ denotes the bond order.

To this end, we propose an algorithm which iteratively updates a record $r(a)$ of available valence for each atom vertex $a \in \mathcal{A}$. The key idea is to (a) update the record accordingly for each addition of atom and bond orders and (b) pre-allocate valence for the `branch_start` and `res_atom` operations by the amount of minimum bond order $\min_{x \in \mathcal{X}_{\texttt{bond}}} o(x)$. The second part (b) is required since the `branch_start` and `res_atom` operations indicate future bond vertices to be added as a neighbor of the current atom vertex. We provide the full description in Algorithm 4.

---

**Algorithm 4** Determination of valence rule violation

---

1: **Input:** Intermediate tree $\mathcal{T} = (\mathcal{A}_{\mathcal{T}}, \mathcal{B}_{\mathcal{T}}, \mathcal{E}_{\mathcal{T}})$, current pointer vertex $i_{\texttt{point}}$, previous pointer vertex $\tilde{i}_{\texttt{point}}$, current decision $d$, previous decision $\tilde{d}$, and record $r(\cdot)$ of available valence.
2: **Output:** Newly updated $r$ and the list $\mathcal{D}$ of decisions that violates the valence rule.
3: **if** $d \in \mathcal{X}_{\texttt{atom}}$ **then**
4:     Set $r(i_{\texttt{point}}) \leftarrow v(d)$.                                       ▷ *Initialize record by atom valence.*
5:     Set $r(i_{\texttt{point}}) \leftarrow r(i_{\texttt{point}}) - o(\tilde{d})$.         ▷ *Update record using previously added bond vertex.*
6:     Set $\mathcal{D} \leftarrow \{x | x \in \mathcal{X}_{\texttt{bond}}, o(x) > r(i_{\texttt{point}})\}$.        ▷ *Reject bond orders higher than the record.*
7:     **if** $r(i_{\texttt{point}}) < \min_{x \in \mathcal{X}_{\texttt{bond}}} o(x)$ **then**
8:         Set $\mathcal{D} \leftarrow \mathcal{D} \cup \{$"(", "*"$\}$.         ▷ *Reject decisions requiring minimal amount of valence.*
9: **if** $d \in \mathcal{X}_{\texttt{bond}}$ **then**
10:     **if** $\tilde{d} \neq$ "(" **then**
11:         Set $r(\tilde{i}_{\texttt{point}}) \leftarrow r(\tilde{i}_{\texttt{point}}) - o(d)$.        ▷ *Update record of previously added atom vertex.*
12:     **else**
13:         Set $r(\tilde{i}_{\texttt{point}}) \leftarrow r(\tilde{i}_{\texttt{point}}) - o(d) + \min_{x \in \mathcal{X}_{\texttt{bond}}} o(x)$.
                                           ▷ *Update previously added atom vertex considering pre-allocated valence.*
14:     Set $\mathcal{D} \leftarrow \{x | x \in \mathcal{X}_{\texttt{atom}}, v(x) < r(i_{\texttt{point}})\}$.        ▷ *Reject atom valence lower than bond order.*
15:     Set $\mathcal{D} \leftarrow \mathcal{D} \cup \{x | x \in \mathcal{L}_{\texttt{res}}, r(x) < o(d) - \min_{x \in \mathcal{X}_{\texttt{bond}}} o(x)\}$.
                                        ▷ *Reject residual edge candidates with valence lower than bond order.*
16: **if** $d =$ "(" **then**
17:     Set $r(i_{\texttt{point}}) \leftarrow r(i_{\texttt{point}}) - \min_{x \in \mathcal{X}_{\texttt{bond}}} o(x)$.        ▷ *Pre-allocate minimum bond order.*
18:     Set $\mathcal{D} \leftarrow \{x | x \in \mathcal{X}_{\texttt{bond}}, o(x) > r(i_{\texttt{point}})\}$.        ▷ *Reject bond orders higher than the record.*
19: **if** $d =$ ")" **then**
20:     Set $\mathcal{D} \leftarrow \emptyset$
21: **if** $d =$ "*" **then**
22:     Set $r(i_{\texttt{point}}) \leftarrow r(i_{\texttt{point}}) - \min_{x \in \mathcal{X}_{\texttt{bond}}} o(x)$.        ▷ *Pre-allocate minimum bond order.*
23:     Set $\mathcal{D} \leftarrow \{x | x \in \mathcal{X}_{\texttt{bond}}, o(x) > r(i_{\texttt{point}})\}$.        ▷ *Reject bond orders higher than the record.*
24: **if** $d \in \mathcal{L}_{\texttt{res}}$ **then**
25:     Set $r(d) \leftarrow r(d) + \min_{x \in \mathcal{X}_{\texttt{bond}}} o(x) - o(x_{i_{\texttt{point}}})$.   ▷ *Update record of previously added atom vertex.*
26:     Set $\mathcal{D} \leftarrow \emptyset$.

---

Given Algorithm 3 and 4, we establish the following theoretical guarantee.

**Definition 2.** *A valid molecular graph* $\mathcal{G} = (\mathcal{A}, \mathcal{B}, \mathcal{E})$ *satisfies the valency rule if* $v(x_a) \geq \sum_{b \in \mathcal{N}(a)} o(x_b)$ *for every atom vertex* $a \in \mathcal{A}$.

**Theorem 2.** *Let* $\mathcal{G} = (\mathcal{A}, \mathcal{B}, \mathcal{E})$ *be a graph being updated by Algorithm 1 and a sequence of decisions* $d_1, \ldots, d_T$. *If the sequence of decisions satisfies the criteria defined by Algorithm 3 and 4, the corresponding graph* $\mathcal{G}$ *satisfies the valency rule.*

*Proof.* To prove the validity of our algorithms, we show that (1) $r(a) \leq v(x_a) - \sum_{b \in \mathcal{N}(a)} o(x_b)$ and (2) $r(a) > 0$ at any time-step applying Algorithm 4 to the sequence of decisions $d_1, \ldots, d_T$.

For (1), we note that a record of an atom $a$ is initialized as $v(x_a) - \sum_{b \in \mathcal{N}(a)} o(x_b)$ whenever it is newly added to the graph $\mathcal{G}$ by a decision. Furthermore, whenever a new edge is added by `attach_bond` and `res_bond` operation, the corresponding bond order $o(x_b)$ is deducted from the record. Importantly, a minimum bond order $\min_{x \in \mathcal{X}_{\text{bond}}} o(x)$ is also added to the record for a `attach_bond` operation consecutive to a `branch_start` operation or a `res_bond` operation. We note how this does not harm (1) since the minimum bond order has already been deducted (or pre-allocated) by the corresponding `branch_start` and `res_atom` operations, respectively.

For the case of (2), one can observe that Algorithm 4 filters out the atoms and bonds that will deduct the record of the corresponding atom to be negative. This completes proves the correctness of our theorem.

$\square$

## D  IMPLEMENTATION DETAILS

In this section, we provide specific details on how we implement the STGG framework for our experiments.

**Training detail.** For all the experiments, we train the Transformer under STGG framework for 100 epochs with batch size of 128 for all the dataset. We use the AdamW (Loshchilov & Hutter, 2019) optimizer with constant learning rate of $10^{-4}$. We use three and six Transformer layers for {QM9, ZINC250K} and MOSES, respectively. The rest of Transformer-related configurations follow that of the original work (Vaswani et al., 2017); we use the attention module with embedding size of 1024 with eight heads, MLP with dimension of 2048, and dropout with probability of 0.1. Using a single Quadro RTX 6000 GPU, it takes approximately three, ten, and 96 hours to fully train the models on QM9, ZINC250K, and MOSES datasets, respectively.

**Pre-processing.** For the whole dataset, we use the following set of atom vocabularies $\mathcal{X}_{\texttt{atom}}$: { `"CH"`, `"CH2"`, `"CH-"`, `"CH2-"`, `"C"`, `"N-"`, `"NH-"`, `"N"`, `"NH"`, `"N+"`, `"NH+"`, `"NH2+"`, `"NH3+"`, `"O-"`, `"O"`, `"O+"`, `"OH+"`, `"F"`, `"P"`, `"PH"`, `"PH2"`, `"P+"`, `"PH+"`, `"S-"`, `"S"`, `"S+"`, `"SH"`, `"SH+"`, `"Cl"`, `"Br"`, `"I"`, }. Note that we assign different features for the same atom numbers with different number of explicit hydrogens and formal charges. This allows our algorithm to properly allocate maximum valence for each atom feature. Next, we use the bond vocabulary $\mathcal{X}_{\texttt{bond}} = \{$ `"-"`, `"="`, `"#"` $\}$, corresponding to bond orders of single, double, and triple, respectively. For explicit calculation of the atom valence during molecular construction, we train our models on kekulized molecules, i.e., aromatic bonds are fixed to single or double bonds.

# E  EXAMPLE OF GENERATED MOLECULES

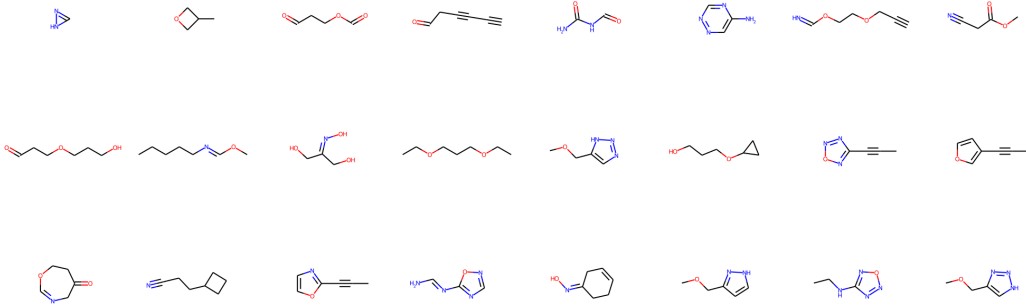

Figure 7: Example of molecules generated from the QM9 dataset.

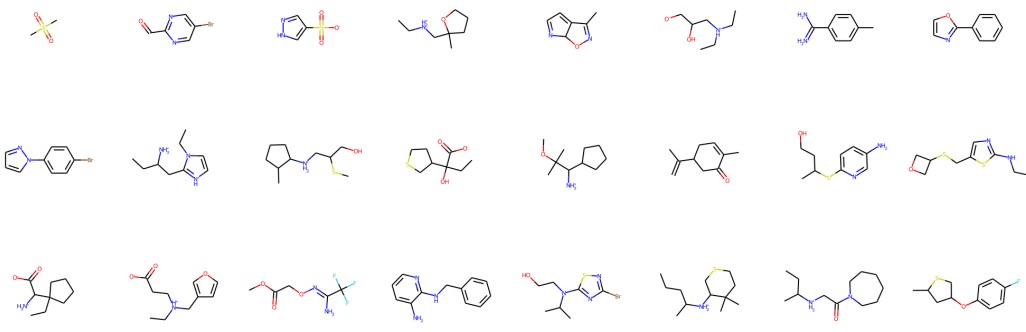

Figure 8: Example of molecules generated from the ZINC250K dataset.

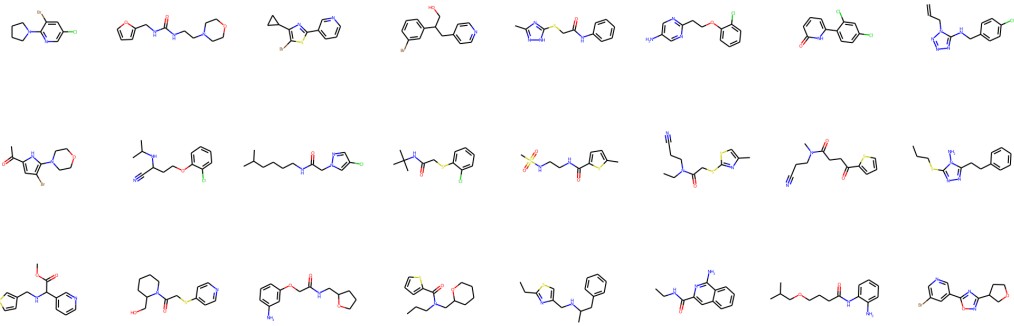

Figure 9: Example of molecules generated from the MOSES dataset.

## F    OFFLINE OPTIMIZATION OF MOLECULES

In this section, we describe our offline molecular optimization algorithm mainly inspired by existing works in offline reinforcement learning (Schmidhuber, 2019; Chen et al., 2021; Janner et al., 2021) and offline model-based optimization (Kumar & Levine, 2020).

For maximizing a reward function defined on a molecule, our algorithm consists of two simple steps. First, our offline optimization algorithm trains a conditional generative model $p_\theta(m|\gamma)$ where $m$ is the molecule and $\gamma$ is the reward function evaluated on the offline dataset of molecules. Next, the reward-conditional generative model samples highly-rewarding molecules by generation conditioned on high values of $\gamma$. In particular, we set the value of $\gamma$ to extrapolate outside the training dataset. Based on the highly expressive power of Transformer architecture, our algorithm can successfully generate highly-rewarding molecules.

# G   ADDITIONAL EXPERIMENTAL RESULTS ON MOLECULAR OPTIMIZATION

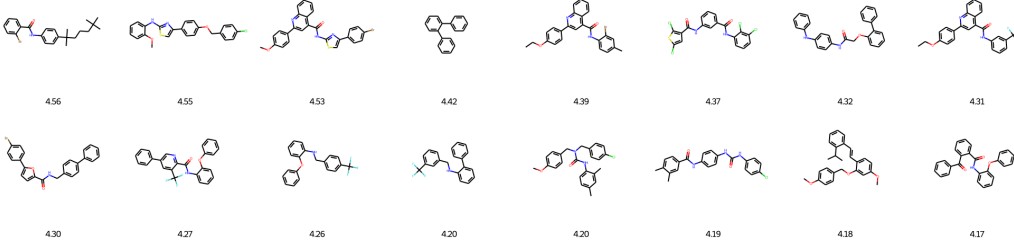

Figure 10: Top-16 molecules generated under condition $\gamma = 4$.

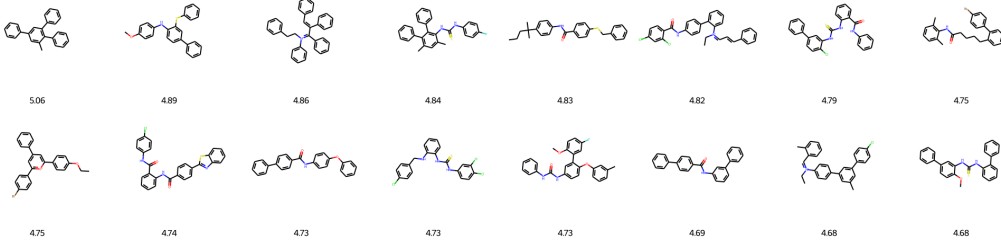

Figure 11: Top-16 molecules generated under condition $\gamma = 5$.

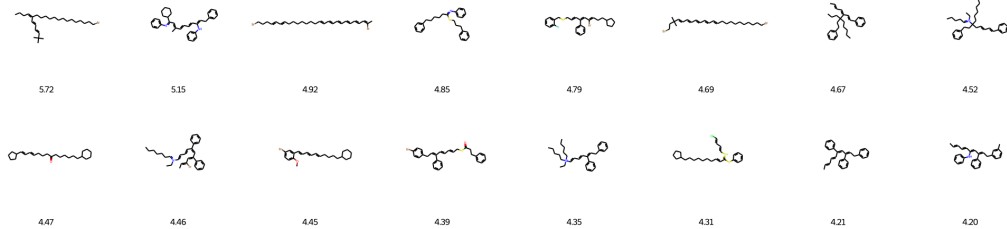

Figure 12: Top-16 molecules generated under condition $\gamma = 6$.

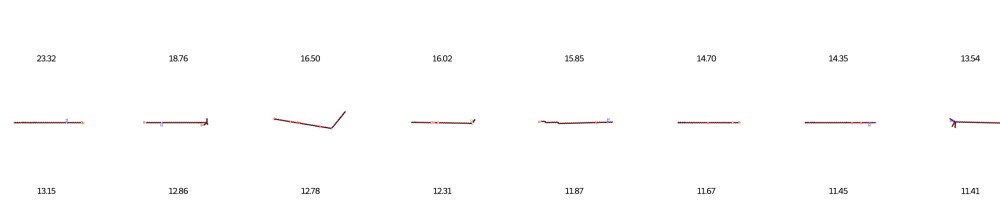

Figure 13: Top-16 molecules generated under condition $\gamma = 7$.

# H    COMPARISON WITH CG-VAE

Table 6: Experimental results for the QM9 benchmark.

| METHOD | VALID | UNIQUE | NOVEL | FCD (↓) | SNN (↑) | FRAG (↑) | SCAF (↑) |
|---|---|---|---|---|---|---|---|
| CG-VAE (Liu et al., 2018) | **1.0000** | **0.9857** | **0.9435** | 1.8515 | 0.3940 | 0.9484 | 0.6628 |
| STGG (ours) | **1.0000** | 0.9561 | 0.6978 | **0.5851** | **0.9998** | **0.9984** | **0.9416** |

Table 7: Experimental results for the ZINC benchmark.

| METHOD | VALID | UNIQUE | NOVEL | FCD (↓) | SNN (↑) | FRAG (↑) | SCAF (↑) |
|---|---|---|---|---|---|---|---|
| CG-VAE (Liu et al., 2018) | **1.0000** | **1.0000** | **0.9982** | 11.335 | 0.2656 | 0.8118 | 0.2411 |
| STGG (ours) | **1.0000** | 0.9996 | 0.9978 | **0.2778** | **0.4664** | **0.9932** | **0.7192** |

In this section, we additionally compare our STGG framework with the CG-VAE model (Liu et al., 2018), which is another atom-by-atom graph generative model that allows masking out the action space to generate molecules satisfying the valence rules. Compared to Table 2, we additionally use the FCD, SNN, Frag, Scaf metrics to measure the faithfulness of the generative models for learning the underlying distribution of molecules. Note that the VALID metric used in Table 2 is insufficient to compare the faithfulness of STGG and CG-VAE since they both enjoy the guarantee to generate molecules satisfying the valence rule.

In Table 6 and 7, one can observe that our algorithm highly outperforms the CG-VAE in terms of faithfully learning the underlying distribution of molecules at the cost of relatively lower UNIQUE. For example, FCD score of our STGG for learning the ZINC dataset is 0.2775 while that of the CG-VAE is 11.33. This highlights the expressive power of our STGG framework.

