# OpenReview forum: "Spanning Tree-based Graph Generation for Molecules"
_ICLR.cc/2022/Conference — ICLR 2022 Spotlight_

### Official Review · Reviewer_4r4T · 2021-10-27

**Correctness:** 4
**Technical Novelty And Significance:** 3
**Empirical Novelty And Significance:** 2
**Recommendation:** 8
**Confidence:** 4

**Main Review:**

## Strengths
### + Reasonable approach to construct a molecule based on spanning-tree representation
While there are a number of approaches to generate a graph based on spanning-tree representation (i.e., SMILES), most of them treat a SMILES representation as a text and do not exploit the spanning tree. The method proposed in this paper explicitly uses the spanning-tree representation to construct a molecule, and it seems a very reasonable approach to construct a molecule, if inspired from SMILES.

### + Clarification on the known flaws in the penalized log P score
I appreciate the authors clarify the limitation of this benchmark task and propose an alternative evaluation scheme. As stated in Section 5.3, this benchmark does not serve as a proper benchmark task, but not many researchers do not discuss it. This statement is very important to the community.


## Weaknesses
### - Relationship to the existing work
I appreciate the authors summarize the relationship to the existing work in Section 4. I understand that the proposed generation procedure is different from the existing ones, but I don't understand why the difference is important. For example, the authors state that "Our STGG framework differs from this line of research since it proposes a new type of graph-based operations for generating the molecular graph", but do not clarify how it is different, and why the difference is important. Such a comparison is important for readers to understand the essence of the proposed method.

### - No theoretical guarantee to generate valid molecules
I am not convinced with the mechanism to comply the valence rule, and I wonder if there is a theoretical guarantee that this mechanism can comply the rule or there is a counter-example where this mechanism cannot guarantee it.
When constructing a ring, it is desirable that the tail atom has one remaining valence, and the ring closes by adding a residual edge. Is it possible that the tail atom has no remaining valence and the ring cannot be closed? For example, C*=C-C≡C seems not to be rejected by the mechanism, but we cannot close the ring.
% This may be a question, rather than weakness. If there is any misunderstanding, please correct it.

### - Relationship to the classical VAE+BO approaches
As discussed in Section 5.3, one of the major issues in the plogP optimization task is that unrealistic molecules can optimize the score. I consider there had been an implicit agreement that the optimized molecules should resemble the training data, which leads to the classical molecular optimization method combining VAE trained on the real-world molecules and Bayesian optimization. As far as I am aware of, the method by Kajino [Kajino, 19] achieves the best scores among the methods using this approach. While the proposed method can control the trade-off between the score and realisticness, it seems the proposed method is not better than VAE+BO approaches in this setting.

[Kajino, 19] Molecular Hypergraph Grammar with Its Application to Molecular Optimization, ICML-19.

---

Given the discussion below, all of my concerns have been addressed.

**Summary Of The Paper:**

This paper presents a method to construct a molecular graph, which is inspired by a spanning tree. In specific, a molecule is constructed by a sequence of actions, each of which adds an atom, adds a bond, starts/ends a branch, adds residual edges for a circular structure, and terminates. The generative process is controlled by a transformer-based neural network, which is specialized to tree construction procedure.
The empirical studies show that the proposed method can learn the distribution of molecules comparably or better than the existing methods, and can be used to molecular optimization.

**Summary Of The Review:**

I am tending to reject this paper because of the following reason.
While the proposed method is very reasonable if inspired by SMILES, its novelty within the literature of graph-based models is less clear, and thus, I am not very enthusiastic about the proposed method. I would increase the score if I were convinced of the contribution of this work to the literature of graph-based models more clearly. So, I would like the authors to clarify which aspect of the proposed graph generation method is important and why. In addition, I would like to ask the authors to discuss why they don't take the VAE+BO approach to molecular optimization, which can implicitly constrain the output to be similar to the training molecules.

---

Nov 21. Updated the score to 5, given the first response by the authors. But still not positive to accept this paper because of the inconsistency in the experiment on molecular optimization.

--

Nov 24. Since all of my concerns have been addressed in the current version, I would recommend to accept this paper.

---

> ### Author Response · Authors · 2021-11-19
> **Reponse to Reviewer 4r4T**
>
> We deeply appreciate your efforts and insightful comments to improve the manuscript. As the reviewers highlighted, our work proposes a novel idea (reviewer xNX3, PBPv) with a reasonable approach (you), validated through comprehensive experiments (reviewer nQAs, xNX3, PBPv). We believe that our STGG framework makes a solid contribution to the literature of molecule-generating deep neural networks.
> In response to the comments, we have carefully revised and enhanced the manuscript, including the following additional discussions and experiments:
> - Improved description for the relationship between our work and the existing SMILES-based and atom-by-atom graph generative models (Section 4)
> - Theoretical guarantee on our framework generating valid molecular graphs (Appendix B).
> - Adding experiments comparing our work to the existing atom-by-atom generative model CG-VAE (Appendix H)
> In the revised manuscript, these updates are temporarily highlighted in "red” for your convenience to check. We think that your review allowed us to improve the manuscript by clarifying our contribution compared to prior works and algorithmic details. Our responses to all your comments are provided below.
>
> **1\. Relationship to the existing works (graph-based generative models)**
>
> Compared to the graph-based generative models, our framework simplifies the decision space for molecule construction by exploiting their tree-like graph structures and using tree-constructive operators. To be specific, our framework requires $O(|\mathcal{A}|+|\mathcal{B}|)$ decisions for constructing a molecule while the existing atom-by-atom graph generative models require $O(|\mathcal{A}|^2)$ decisions. This implies that our generative model requires a smaller number of decisions for sparse graphs like molecules, i.e., when $|\mathcal{B}|$ is small. Such a simplification allows our STGG to learn the distribution of molecules more faithfully than the existing atom-by-atom graph generative models. Finally, we also remark that our work is the first to successfully use the Transformer architecture as a graph-based generative model. We have added the respective discussion in Section 4 of our manuscript.
>
> **2\. No theoretical guarantee to generate valid molecules**
>
> To incorporate your comment, in Appendix B, we provide a theoretical guarantee of our algorithm generating valid molecules. As an additional empirical validation, we tried randomly generating 10,000 molecules using our algorithm and the generated molecules were 100% valid. Regarding your comment on open rings, we note that a sequence of decisions with open rings may still define a valid molecular graph by ignoring the open ring. Your example of “C*=C-C≡C” will generate a valid molecule corresponding to the sequence of decisions “C=C-C≡C”.
>
> **3\. Relationship to the classical VAE + BO approaches (underperformance and justification)**
>
> First, we would like to clarify that our offline optimization algorithm does not underperform compared to the VAE + BO approach. The suggested VAE+BO baseline, i.e., [Kajino, 2019], achieves the PLogP score of 5.24 while our STGG achieves 27.84.
>
> We believe that our algorithm is better than the VAE+BO in terms of easily controlling the tradeoff between realistic-ness and the optimization objective based on tuning the value of $\lambda$. Indeed, while we agree that the VAE+BO approach is attractive, it can still suffer from the accumulated feedback of an ill-defined scoring function without proper tuning, e.g., see the molecules proposed by [Tripp et al., 2020]. Namely, the acquisition function of BO often encourages querying for molecules outside the distribution using its exploration mechanism. When the BO is not properly regularized, it may query for unrealistic molecules and include it into the training dataset. Properly regularizing the VAE+BO approach is an active area of research [Griffith et al., 2020, Notin et al., 2021] and we believe our offline optimization algorithm could serve as an alternative paradigm for optimizing realistic molecules.
>
> [Tripp et al., 2020] Sample-Efficient Optimization in the Latent Space of Deep Generative Models via Weighted Retraining
>
> [Griffith et al., 2020]  Constrained Bayesian optimization for automatic chemical design using variational autoencoders
>
> [Notin et al., 2021] Improving black-box optimization in VAE latent space using decoder uncertainty

---

> > ### Comment · Reviewer_4r4T · 2021-11-20
> > **Re: Reponse to Reviewer 4r4T**
> >
> > I would appreciate the authors to sincerely respond to my review. The first and concerns are partially addressed, but the revised paper is not very satisfactory. The third one is not addressed. Given these results, I would like to raise my score to 5. Since the experiment on molecular optimization is not consistent and thus convincing, I am still not positive to accept this paper.
> >
> > ## 1. Relationship to the existing works (graph-based generative models)
> > I am partially convinced of the contribution of this work to the literature of graph-based generative models.
> > While I am very convinced of the relationship to the atom-by-atom generative models in terms of the number of decisions, I am not very convinced of the relationship to the grammar-based generative models, because the latter models can generate a molecule by a fewer number of decisions. The authors instead suggest the benefit of the proposed method may be that it can learn the inner semantics of substructures whereas a method like JT-VAE cannot, but this is not well studied in the experiments. I would suggest the authors to verify this conjecture in the experiments.
> >
> > ## 2. No theoretical guarantee to generate valid molecules
> > First of all, I'm sorry that the example I was using “C*=C-C≡C” was wrong and what I meant is that an intermediate sequence of decisions like “C*=C-C≡N” has no valid decision, because N already has three bonds. But given the response by the authors, I understand the only valid decision is [eos], and the sequence can be safely converted into a valid molecule.
> >
> > I understand that practically the proposed method can generate a valid molecule, but it seems there are several important points that are not written explicitly in the paper. First, the terminate operation [eos] is allowed even when the stack and the list are not empty (which is explained in the updated Algorithm 3, but the main text says that "the terminate operation is only allowed when the stack S_branch and the list L_res is empty."). Second, a sequence of decisions with open rings (e.g., C-C*-C[eos]) may be safely converted into a valid molecule, but the procedure is not clarified. These two points, although they are not very beautiful, should be discussed in the main body.
> >
> > It seems Theorem 1 states that the proposed algorithm can generate a valid sequence of decisions, which does not necessarily lead to a valid molecule (e.g., it may violate the valency condition). I would like to see a definition of a valid molecule (like "a valid molecular graph is a graph with the following properties...") and a theorem like "the proposed algorithm always generates a valid molecular graph".
> >
> > ## 3. Relationship to the classical VAE + BO approaches (underperformance and justification)
> > Given the discussion on the ill-definedness of the penalized logP score, it is not convincing to evaluate the optimization performance just by the score, but it is quite necessary to consider the "realisticness" of the optimized molecules.
> > Let me temporally define the realisticness by whether the optimized molecules have chain-like structures or not (although this is very qualitative and we definitely needs quantitative metrics...). Then, the top-3 molecules discovered by the method by Kajino have scores 5.56, 5.40, and 5.34, which have no chain-like structures and thus are realistic, while the realistic molecules discovered by the proposed method have scores equal to or less than 5.15, which suggests that the proposed method could not discover realistic molecules with better penalized log P scores.
> > In summary, I find inconsistency in that, while this experiment is motivated by the ill-definedness of the penalized log P score and the authors tried to alleviate it by using an offline method, the proposed method and the others are compared purely in terms of the score, without taking the realisticness into account. And if taking it into account in a very naive way, it is likely that the proposed method is no better than the method by Kajino.

---

> > > ### Author Response · Authors · 2021-11-22
> > > **Re: Re: Reponse to Reviewer 4r4T**
> > >
> > > Thank you very much for your quick response and great suggestions to improve our paper, we do not take it for granted. We are very happy to hear that our rebuttal was helpful and we have made further updates to incorporate your additional feedback.
> > >
> > > **1. Relationship to the existing works (graph-based generative models)**
> > >
> > > We would like to further alleviate your concern on our relationship to grammar-based generative models, i.e., how our STGG can "learn inner semantics of substructures." We believe such a hypothesis can be verified by showing that our method (a) outperforms the grammar-based for learning distribution of the molecules and (b) can generate substructures unseen in the training dataset of molecules. We note how we showed (a) in our experiments using the MOSES benchmark. Furthermore, (b) is trivial since our STGG assigns non-zero probability even to substructures unseen in the training dataset of molecules.
> > >
> > > **2. No theoretical guarantee to generate valid molecules**
> > >
> > > Thank you for the helpful suggestions to make our paper more clear and transparent. We have become aware of our mistake of stating "the terminate operation is only allowed ..." and incorporated your comments to remove the editorial flaw. We have also updated the appendix to include definitions and theorems as you have requested.
> > >
> > > **3. Relationship to the classical VAE + BO approaches (underperformance and justification)**
> > >
> > > We mainly designed our molecular optimization experiment to yield a consistent message: our STGG is useful for optimizing molecules. We show this by using STGG to (a) solely maximize the PLogP score and (b) maximize the PLogP score under the realistic-ness constraint. We do not evaluate our method just by the score, i.e., (a), but also demonstrate STGG generating realistic molecules with relatively lower scores, i.e., (b). As you have mentioned, it is non-trivial to define a quantitative metric of "realistic-ness" for (b) and we do not compare to the existing methods for (b) by some scoring function.
> > >
> > > Despite our argument on ill-definedness of the PLogP score, we still think (a) is important since it demonstrates the ability of STGG to learn a narrow distribution of high-scoring molecules. Furthermore, the existing experiments on graph generative models are tuned for (a) and we think it is unfair to compare to their results in terms of another metric, i.e., (b). To consider this aspects, we aimed to design a experiment showing that our STGG can achieve good results for both worlds.
> > >
> > > Therefore, while we agree that our offline algorithm does not show SOTA performance for some quantitative metric in molecular optimization, we do think that it offers a very solid and unique merit to the literature. Our offline algorithm also enjoys the benefit of not requiring any expensive queries to evaluate the molecules (in contrast to [Kajito, 2019]). We have revised Section 5.3 to make this point clear and emphasize your point on how experiments should not compare purely in terms of achieving high scores. We also added [Kajito, 2019] as a baseline of our experiment.

---

> > > > ### Comment · Reviewer_4r4T · 2021-11-22
> > > > **Re: Re: Re: Reponse to Reviewer 4r4T**
> > > >
> > > > Thanks for quick updates.
> > > >
> > > > __3. Relationship to the classical VAE + BO approaches (underperformance and justification)__
> > > >
> > > > I'm still not convinced of this setting. I consider there are at least two options for those who consider the penalized log P task is not appropriate.
> > > >
> > > > 1. Use the goal-directed benchmarks of GuacaMol
> > > > 2. Use a predictor of the target property trained on real-world data (e.g., "Multi-Objective Molecule Generation using Interpretable Substructures" by Jin et al.)
> > > >
> > > > I really appreciate the authors develop an algorithm to control the score and realistic-ness of molecules, but without a quantitative measure of realistic-ness, the performance of such an algorithm cannot be properly evaluated; in fact, the authors could compare the performance of the proposed method only in the non-realistic mode, but could not reasonably study its performance in a realistic mode, failing to avoid from the ill-defined benchmark.
> > > >
> > > > I like the idea of controlling the trade-off, but in order to argue the benefit of such an algorithm, it is mandatory to develop a measure of realistic-ness; otherwise, it is difficult to scientifically understand its benefit.
> > > >
> > > > I don't know if it is necessary to cite the scores of [Kajino 2019] (and some of the scores in Fig. 5), because the authors did not mention them in the main text. I suggest to cite papers _only if_ they are necessary. Also, if the scores have to be cited, they should be 5.56, 5.40, 5.34 (see Table 1 [Kajino, 2019]), which are obtained in the same setting as [Jin+, 2018].

---

> > > > > ### Author Response · Authors · 2021-11-23
> > > > > **Re: Re: Re: Re: Reponse to Reviewer 4r4T**
> > > > >
> > > > > We deeply appreciate your frequent communication with us. We updated the revised the paper to incorporate your comment on citations to [Kajino 2019]. We included it in the comparison since it is an existing graph generative neural network with high performance of PLogP optimization (while retaining the realistic-ness).
> > > > >
> > > > > We sincerely disagree that the proposed GuacaMol benchmark and the target property predictor used by [Jin et al., 2020] bypass the caveat of penalized log P task. The suggested scoring function also does not consider realistic-ness of the molecules and may assign high scores to unstable, unsynthesizable, and toxic molecules.
> > > > >
> > > > > Next, we thank your appreciation of our algorithm and we agree that newly developing metrics to measure realistic-ness of molecules would strengthen our empirical validation. However, we also believe that the current experiments sufficiently show the useful-ness of our STGG algorithm. Even without a measure of realistic-ness, one can simply observe the molecules generated by our algorithm and verify our claim that "our algorithm can generate realistic molecules with relatively high scores". This is also how the existing VAE+BO approaches have shown their optimized molecules to resemble the training dataset.
> > > > >
> > > > > We think the development of a new realistic-ness metric deserves a separate study since (a) our work is not mainly focusing on the molecular optimization task and (b) it is a significant problem that requires thorough investigation.

---

> > > > > > ### Comment · Reviewer_4r4T · 2021-11-23
> > > > > > **Re: Re: Re: Re: Re: Reponse to Reviewer 4r4T**
> > > > > >
> > > > > > Thanks for clarifications. Since I like the method itself and the remaining concern is how to present the results of the molecular optimization task, I am not against accepting this paper if the authors clarify in the paper that (i) the main message of the molecular optimization task is to illustrate the capability of the proposed method to control the trade-off between the score and realisticness, not to demonstrate that the proposed method can generate realistic molecules better than other methods and (ii) it is necessary to develop a measure of realisticness in order to compare the performance of different methods.

---

> > > > > > > ### Author Response · Authors · 2021-11-23
> > > > > > > **Re: Re: Re: Re: Re: Re: Reponse to Reviewer 4r4T**
> > > > > > >
> > > > > > > Thank you for the great suggestions. We fully agree with your points and incorporated your comments in our updated experimental section.
> > > > > > >
> > > > > > > 1. As the main message of our paper, we state: "Using this algorithm, we demonstrate how our STGG is capable of generating both (a) high-scoring molecules and (b) realistic molecules with a reasonably high score."
> > > > > > >
> > > > > > > 2. We clearly state that our method does not strictly outperform the baselines: "However, we also remark that our results do not imply our optimization results to be strictly better than the baselines; we believe it is necessary to develop and incorporate quantitative measures for realistic-ness of molecules to fairly evaluate the molecular optimization algorithms. We believe such a research to be a important future direction."

---

> > > > > > > > ### Comment · Reviewer_4r4T · 2021-11-23
> > > > > > > > **Re: Re: Re: Re: Re: Re: Re: Reponse to Reviewer 4r4T**
> > > > > > > >
> > > > > > > > Thanks for the update. I confirm that all of my concerns are addressed in the updated manuscript, and thus, I would like to recommend to accept this paper. I really appreciate the authors' constructive attitude to this discussion.
> > > > > > > >
> > > > > > > > Typo:
> > > > > > > > p.9 "a important future dirction" -> "an important future direction"

---

> > > > > > > > > ### Author Response · Authors · 2021-11-24
> > > > > > > > > **Re: Re: Re: Re: Re: Re: Re: Re: Reponse to Reviewer 4r4T**
> > > > > > > > >
> > > > > > > > > Thank you very much for the positive response. We think your response were very helpful to improve your paper, and we are very happy to hear that your concerns are addressed!

---

> ### Comment · Area_Chair_e15m · 2021-11-20
> **Reviewer 4r4T**
>
> Reviewer 4r4T: Your recommendation is the most negative one among all reviewers. Authors have provided detailed response to your questions. Could you please read and respond to these answers?

---

### Official Review · Reviewer_PBPv · 2021-10-30

**Correctness:** 3
**Technical Novelty And Significance:** 3
**Empirical Novelty And Significance:** 4
**Recommendation:** 8
**Confidence:** 4

**Main Review:**

Strengths:
I think this work has a lot of novelty and merit. The idea of using a transformer which can view earlier decisions with attention mechanisms increases the possibility of valid molecular generation. This also forms a conceptual departure from highly sequential or reinforcement learning-based methods. They show their results on 3 datasets MOSES, QM9 as well as ZINC250 K. As can be seen in the results, many previous methods do not run well on QM9 and are often overfit for 1-2 benchmarks. The authors also perform ablation tests which show the improvements with several of their design choices including the use of graph structure, the use of a transformer and the use of graph construction and valence masks.

Weaknesses:
Although the ablation studies that are used by the author validate some of the design decisions they don't validate or establish the need for several of their most novel decisions: 1. The two-pronged system of first generating spanning trees and then residual edges, 2. the use of a bipartite graph rather than one that uses edges to represent bonds, 3. the use of spanning trees themselves. In addition I would like to see some results on how long it takes to train this transformer model as opposed to other models, due to what I believe is a highly increased number of parameters given the positional encoding and the attention mechanisms.





**Summary Of The Paper:**

This paper proposes a new paradigm of graph generation, a transformer network sequentially generates a sequence of decisions to generate a spanning tree for a bipartite graph. These decisions have 7 forms such as attach atom, attach bond, branch start,  etc. After the spanning tree, they add residual edges. The authors also have a focus on generating valid graphs (which is seen in the results) in which they mask out invalid decisions during the generation process itself.



**Summary Of The Review:**

Interesting proposal, needs more validation.

---

> ### Author Response · Authors · 2021-11-19
> **Response to Reviewer PBPv**
>
> We deeply appreciate your efforts and insightful comments to improve the manuscript. As the reviewers highlighted, our work proposes a novel idea (you and reviewer xNX3) with a reasonable approach (reviewer 4r4T), validated through comprehensive experiments (you, reviewer nQAs, and reviewer xNX3). We believe that our STGG framework makes a solid contribution to the literature of molecule-generating deep neural networks.
> In response to the comments, we have carefully revised and enhanced the manuscript, including the following additional discussions and experiments:
> - Improved description for the relationship between our work and the existing SMILES-based and atom-by-atom graph generative models (Section 4)
> - Theoretical guarantee on our framework generating valid molecular graphs (Appendix B).
> - Adding experiments comparing our work to the existing atom-by-atom generative model CG-VAE (Appendix H)
> In the revised manuscript, these updates are temporarily highlighted in "red” for your convenience to check. We think that your review allowed us to improve the manuscript by clarifying the effectiveness of our algorithmic components and providing more comprehensive experimental details. Our responses to all your comments are provided below.
>
> **1\. Validating and establishing the need of design choices**
>
> Thank you for the constructive suggestion. We establish the need of the remarked design choices as follows:
> - **Two-pronged system of first generating spanning trees and then residual edges.** The two-pronged system of (1) generating spanning trees and (2) residual edges is required for decomposing a molecular graph into (1) a subgraph that can be generated using tree-constructive operators and (2) the remaining edges constructed by residual edge operations. This allows generating a large portion of the given molecular graph using compact tree constructive operations.
> - **Use of a bipartite graph rather than one that uses edges to represent bonds.** Using the bipartite graph representation of molecules is mainly required for assessing the “position” of the bonds in the given molecular graph. Without using the bipartite graph representation, it is impossible to (1) assign a pointer vertex to the bond of interest and (2) define positional embedding of the bond using tree-based relative positions.
> - **Use of spanning trees.** Since our main idea is to use compact tree-constructive operators over existing graph-constructive operators, we design our framework to use the tree-constructive operators as much as possible. Constructing the spanning tree of the molecule using such a tree-constructive operator is natural since it is a maximum-sized tree in the given molecular graph.
> We also believe that the experiments of our STGG outperforming the existing graph generative models serve as an empirical validation of our design choices. We found that precisely ablating the effectiveness of the design choices listed above is non-trivial since they are the main component of STGG and cannot be replaced by another algorithmic component.
>
> **2\. Training time of the transformer model**
>
> Our model took approximately 12 hours to reach the minimum validation loss on the ZINC dataset using a Tesla V100 core and 8 CPUs. This is much faster than JT-VAE (24 hours) and slower than GCPN (8 hours) and GraphAF (4 hours) for being trained using a Tesla V100 core and 32 CPUs [Shi et al., 2019]. As the reviewer pointed out, the slower running time is likely due to the increased number of parameters. We have incorporated this information in our manuscript.
> [Shi et al., 2019] GraphAF: a Flow-based Autoregressive Model for Molecular Graph Generation

---

### Official Review · Reviewer_xNX3 · 2021-10-31

**Correctness:** 3
**Technical Novelty And Significance:** 3
**Empirical Novelty And Significance:** 3
**Recommendation:** 6
**Confidence:** 5

**Main Review:**

**Strengths**
* The proposed method is quite novel. The key idea is to represent the molecular graph as a sequence of decisions according to a **novel spanning tree-based grammar**. Compared to SMILES-based molecular generative methods, STGG allows inferring the intermediate graph structures and takes the graph structure into consideration. The method is general and can be applied to other graph structures.
* The paper is well written. The experiments of the work are comprehensive and well designed. In particular, for PLOGP molecular optimization, they adopt an offline optimization algorithm, which is interesting to me.
* The code is already provided in the Supplementary Material

**Weaknesses or Questions**
* What is the main advantage of STGG over atom-by-atom graph generative models? The Table2 indicates that STGG tend to generate molecules with low novelty and the main advantage is validity.
The authors mention that atom-by-atom methods determine the validity via sample-reject. But actually, we can also mask out invalid decisions for atom-by-atom methods by keeping a record of valency and bond information.
* **Concerns on training data**: To get the training data, the authors use algorithm2 presented in Appendix A to obtain the sequence of decisions from graph structures. For each graph structure, how many sequences do you get? Do you fix the random seed of this algorithm, or generate the sequence on the fly in each batch with different randomness? My key point is that if the training sequence is generated on the fly for each batch with random DFS, the training data for SMILES-TRANSFORMER should also be augmented (instead of using canonical smiles) for a fair comparison in Table 2.

**Summary Of The Paper:**

The paper proposes a novel spanning tree-based graph generation (STGG) framework. The key idea is to represent the molecular graph as a sequence of decisions according to a **novel spanning tree-based grammar**, and then model these decision sequences using the transformer.

To accommodate such a novel algorithm, several interesting techniques are involved, e.g., representing molecules as bipartite graphs, using tree-based positional encoding for transformer. The authors claim that the proposed method allows generating valid molecular structures and inferring the intermediate graph structure.

**Summary Of The Review:**

I think the paper reaches the bar of ICLR for the reasons listed above.
However, several concerns remain to be addressed.

---

> ### Author Response · Authors · 2021-11-19
> **Response to Reviewer xNX3**
>
> We deeply appreciate your efforts and insightful comments to improve the manuscript. As the reviewers highlighted, our work proposes a novel idea (you and reviewer PBPv) with a reasonable approach (reviewer 4r4T), validated through comprehensive experiments (you, reviewer nQAs, and reviewer PBPv). We believe that our STGG framework makes a solid contribution to the literature of molecule-generating deep neural networks.
> In response to the comments, we have carefully revised and enhanced the manuscript, including the following additional discussions and experiments:
> - Improved description for the relationship between our work and the existing SMILES-based and atom-by-atom graph generative models (Section 4)
> - Theoretical guarantee on our framework generating valid molecular graphs (Appendix B).
> - Adding experiments comparing our work to the existing atom-by-atom generative model CG-VAE (Appendix H)
> In the revised manuscript, these updates are temporarily highlighted in "red” for your convenience to check. We think that your review allowed us to improve the manuscript by clarifying our contribution compared to prior works and experimental details. Our responses to all your comments are provided below.
>
> **1\. Advantage of our algorithm over atom-by-atom graph generative models**
>
> We believe the main advantage of our framework to be its ability to faithfully learn the distribution of molecules. Such a strength is confirmed by the evaluations using metrics like Valid, FCD, SNN, Frag, and Scaf in our experiments. This is different from just achieving a high Valid score, since the existing atom-by-atom graph generative models may learn a distribution far away from the training dataset even after masking out valencewise-invalid decisions. Indeed, in our newly conducted experiment in Appendix H, one can observe how STGG outperforms the existing atom-by-atom graph generative model (which masks out invalid decisions) in terms of accurately learning the distribution of molecules.
>
> Conceptually, our framework simplifies the decision space of atom-by-atom generative models by exploiting the tree-like graph structures of molecules. To be specific, STGG requires $O(|\mathcal{A}|+|\mathcal{B}|)$ decisions for constructing a molecule while the existing atom-by-atom graph generative models, e.g., GraphAF [Shi et al., 2020], require $O(|\mathcal{A}|^2)$ decisions. This implies that our generative model requires a much smaller number of decisions for sparse graphs like molecules, i.e., when $|\mathcal{B}|$ is small. We have added the respective discussion in Section 4 of our manuscript.
>
> [Shi et al., 2020] GraphAF: a Flow-based Autoregressive Model for Molecular Graph Generation
>
> **2\. Concerns on generating the training sequence on fly**
>
> Our comparison is fair since we fix the random seed to construct the sequence of decisions from graph structures similar to how a canonical SMILES representation is obtained from a molecule. This design choice is based on our observation that such a randomization degrades the performance of both STGG and SMILES-based transformer in our experiments due to underfitting.

---

### Official Review · Reviewer_nQAs · 2021-11-02

**Correctness:** 3
**Technical Novelty And Significance:** 3
**Empirical Novelty And Significance:** Not applicable
**Recommendation:** 6
**Confidence:** 4

**Main Review:**

Strength:
 * Comprehensive evaluation. The method is evaluated on standard ZINC250K, QM9, MOSES benchmarks and compared with many previous baselines. The model is able to achieve state-of-the-art results on most of the metrics.

Weakness:
 * As noted in the paper, SMILES strings are also constructed by spanning tree algorithms. As shown in Figure 2, there is very little difference between generating a spanning tree versus generating a SMILES string. The only difference between STGG and SMILES string is the adoption of a tree-based transformer.
 * STGG is a node-by-node graph generation method following the depth-first order (DFS) of a graph. CG-VAE (Liu et al. 2018) is also a node-by-node graph generator, but following the breadth-first order (BFS). It would be great if authors can compare STGG with CG-VAE given their similarity.

**Summary Of The Paper:**

This paper proposes a spanning tree based generative model (STGG) for molecular graphs. STGG sequentially generate a molecule's spanning tree and fill in the residual edges on the way. The spanning tree construction is similar to the standard SMILES representation, but the model operates on a molecular graph rather than a SMILES string. STGG adopts a tree-based transformer with relative positional encoding for tree generation, and a attention-based predictor for residual edge prediction. The method is evaluated on standard ZINC250K, QM9, and MOSES benchmarks and outperform existing baselines.

**Summary Of The Review:**

This paper has different pros and cons. I vote for weak accept because the empirical results seem convincing. Adding comparison with CG-VAE is important and I hope the authors can address this concern during rebuttal.

---

> ### Author Response · Authors · 2021-11-19
> **Response to Reviewer nQAs**
>
> We deeply appreciate your efforts and insightful comments to improve the manuscript. As the reviewers highlighted, our work proposes a novel idea (reviewer xNX3, PBPv) with a reasonable approach (reviewer 4r4T), validated through comprehensive experiments (you, reviewer xNX3, and reviewer PBPv). We believe that our STGG framework makes a solid contribution to the literature of molecule-generating deep neural networks.
> In response to the comments, we have carefully revised and enhanced the manuscript, including the following additional discussions and experiments:
>
> - Improved description for the relationship between our work and the existing SMILES-based and atom-by-atom graph generative models (Section 4)
> - Theoretical guarantee on our framework generating valid molecular graphs (Appendix B).
> - Adding experiments comparing our work to the existing atom-by-atom generative model CG-VAE (Appendix H)
>
> In the revised manuscript, these updates are temporarily highlighted in "red” for your convenience to check. We think that your review allowed us to improve the manuscript by clarifying the contribution of our work compared to prior works and strengthening the empirical validation. Our responses to all your comments are provided below.
>
> **1\. Little difference between STGG and SMILES-based generation.**
>
> We believe there exists a significant difference between the structure-aware STGG and linear SMILES-based generation; STGG allows realizing the intermediate structure of the molecule being constructed while SMILES-based generation cannot. This critical difference allows the adoption of structure-aware deep neural networks (inapplicable to SMILES-based generation) to STGG.
>
> To be specific, the difference between STGG and SMILES-based generation comes from our newly introduced concept of a pointer vertex $i_{\mathtt{point}}$, a vertex-list $\mathcal{L}$, and a vertex-stack $\mathcal{S}$. They allow (1) recognizing an incomplete sequence of decisions as a graph and (2) assigning positions to each decision. In contrast, an incomplete SMILES string (1) does not define a graph structure and (2) assigning positions to each character is non-trivial. We have clarified this point in Section 4 of our manuscript.
>
> **2\. Comparison to CG-VAE**
>
> We agree that CG-VAE is an important baseline and we have incorporated your comment by adding comparison to the CG-VAE baseline in Appendix H of our manuscript. In the new experiment, we compare the quality of molecules generated from our STGG algorithm and the CG-VAE baseline using the QM9 and the ZINC datasets. Since both our STGG algorithm and the CG-VAE baseline guaranteedly generate valid molecules, i.e., achieve 100% validity, we additionally use the FCD, SNN, Frag, Scaf metrics (from the MOSES benchmark) to measure the faithfulness of the generative models for learning the underlying distribution of molecules.
>
>   Table 1: Comparing CG-VAE and STGG on the QM9 dataset
>
> | Method |    Valid   |   Unique   |    Novel   | FCD (&#8595;) | SNN (&#8593;) | Frag (&#8593;) | Scaf (&#8593;) |
> |:------:|:----------:|:----------:|:----------:|:-------------:|:-------------:|:--------------:|:--------------:|
>  | CG-VAE | **1.0000** | **0.9857** | **0.9435** |     1.8515    |     0.3940    |     0.9484     |     0.6628     |
> |  STGG  | **1.0000** |   0.9561   |   0.6978   |   **0.5851**  |   **0.9998**  |   **0.9984**   |   **0.9416**   |
>
>   Table 2: Comparing CG-VAE and STGG on the QM9 dataset
>
> | Method |    Valid   |   Unique   |    Novel   | FCD (&#8595;) | SNN (&#8593;) | Frag (&#8593;) | Scaf (&#8593;) |
> |:------:|:----------:|:----------:|:----------:|:-------------:|:-------------:|:--------------:|:--------------:|
>  | CG-VAE | **1.0000** | **1.0000** | **0.9982** |     11.335    |     0.2656    |     0.8118     |     0.2411     |
> |  STGG  | **1.0000** |   0.9996   |   0.9978   |   **0.2778**  |   **0.4664**  |   **0.9932**   |   **0.7192**   |
>
>   In Table 1 and 2, one can observe that our STGG highly outperforms the CG-VAE in terms of the FCD, SNN, Frag, and Scaf metrics. For example, in Table 2, the FCD score of our STGG for learning the ZINC dataset is 0.2775 while that of the CG-VAE is 11.33. This highlights the power of our STGG framework for faithfully learning the distribution of molecules.

---

> > ### Comment · Reviewer_nQAs · 2021-12-02
> > **Thank you for your response**
> >
> > I have read your response and I will keep my original score.

---

### Decision · Program_Chairs · 2022-01-20

**Decision:**

Accept (Spotlight)

**Comment:**

This paper proposes a spanning tree-based graph generation framework for molecular graph generation, which is an interesting problem. The tree-based approach is efficient and relatively effective in molecular graph generation tasks, and the empirical results are convincing. There were some concerns during the initial reviews, but all of them have been addressed during the discussion phase. Thus, I recommend this work be accepted.